# Online PAC-Bayes Learning

**Maxime Haddouche**
Inria and University College London
France and UK

**Benjamin Guedj**
Inria and University College London
France and UK

## Abstract

Most PAC-Bayesian bounds hold in the batch learning setting where data is collected at once, prior to inference or prediction. This somewhat departs from many contemporary learning problems where data streams are collected and the algorithms must dynamically adjust. We prove new PAC-Bayesian bounds in this online learning framework, leveraging an updated definition of regret, and we revisit classical PAC-Bayesian results with a batch-to-online conversion, extending their remit to the case of dependent data. Our results hold for bounded losses, potentially *non-convex*, paving the way to promising developments in online learning.

## 1 Introduction

Batch learning is somewhat the dominant learning paradigm in which we aim to design the best predictor by collecting a training dataset which is then used for inference or prediction. Classical algorithms such as SVMs [see Cristianini et al., 2000, among many others] or feedforward neural networks [Svozil et al., 1997] are popular examples of efficient batch learning. While the mathematics of batch learning constitute a vivid and well understood research field, in practice this might not be aligned with the way practitionners collect data, which can be sequential when too much information is available at a given time (*e.g.* the number of micro-transactions made in finance on a daily basis). Indeed batch learning is not designed to properly handle dynamic systems.

Online learning (OL) [Zinkevich, 2003, Shalev-Shwartz, 2012, Hazan, 2016] fills this gap by treating data as a continuous stream with a potentially changing learning goal. OL has been studied with convex optimisation tools and the celebrated notion of regret which measures the discrepancy between the cumulative sum of losses for a specific algorithm at each datum and the optimal strategy. It led to many fruitful results comparing the efficiency of prediction for optimisation algorithms such that Online Gradient Descent (OGD), Online Newton Step through static regret [Zinkevich, 2003, Hazan et al., 2007]. OL is flexible enough to incorporate external expert advice onto classical algorithms with the optimistic point of view that such advices are useful for training [Rakhlin and Sridharan, 2013a,b] and then having optimistic regret bounds. Modern extensions also allow to compare to moving strategies through dynamic regret [see e.g. Yang et al., 2016, Zhang et al., 2018, Zhao et al., 2020]. However, this notion of regret has been challenged recently: for instance, Wintenberger [2021] chose to control an expected cumulative loss through PAC inequalities in order to deal with the case of stochastic loss functions.

Statements holding with arbitrarily large probability are widely used in learning and especially within the PAC-Bayes theory. Since its emergence in the late 90s, the PAC-Bayes theory (see the seminal works of Shawe-Taylor and Williamson, 1997, McAllester, 1998, 1999 and the recent surveys by Guedj, 2019, Alquier, 2021) has been a powerful tool to obtain generalisation bounds and to derive efficient learning algorithms. Classical PAC-Bayes generalisation bounds help to understand how a learning algorithm may perform on future similar batches of data. More precisely, PAC-Bayes learning exploits the Bayesian paradigm of explaining a learning problem through a meaningful distribution over a space of candidate predictors [see e.g. Maurer, 2004, Catoni, 2007, Tolstikhin and Seldin, 2013, Mhammedi et al., 2019]. An active line of research in PAC-Bayes learning is to

36th Conference on Neural Information Processing Systems (NeurIPS 2022).

overcome classical assumptions such that data-free prior, bounded loss, iid data [see Lever et al., 2010, 2013, Alquier and Guedj, 2018, Holland, 2019, Rivasplata et al., 2020, Haddouche et al., 2021, Guedj and Pujol, 2021] while remaining in a batch learning sprit. Finally, a pioneering line of work led by [Seldin et al., 2012a,b] on PAC-Bayes learning for martingales and independently developed by [Gerchinovitz, 2011, Foster et al., 2015, Li et al., 2018] boosted PAC-Bayes learning by providing sparsity regret bound, adaptive regret bounds and online algorithms for clustering.

**Our contributions.** Our goal is to provide a general online framework for PAC-Bayesian learning. Our main contribution (Thm. 2.3 in Sec. 2) is a general bound which is then used to derive several online PAC-Bayesian results (as developed in Secs. 3 and 4). More specifically, we derive two types of bounds, *online PAC-Bayesian training and test bounds.* Training bounds exhibit online procedures while the test bound provide efficiency guarantees. We propose then several algorithms with their associated training and test bounds as well as a short series of experiments to evaluate the consistency of our online PAC-Bayesian approach. Our efficiency criterion is not the classical regret but an expected cumulative loss close to the one of Wintenberger [2021]. More precisely, Sec. 3 propose a stable yet time-consuming Gibbs-based algorithm, while Sec. 4 proposes time efficient yet volatile algorithms. We emphasize that our PAC-Bayesian results only require a bounded loss to hold: no assumption is made on the data distribution, priors can be data-dependent and we do not require any convexity assumption on the loss, as commonly assumed in the OL framework.

**Outline.** Sec. 2 introduces the theoretical framework as well as our main result. Sec. 3 presents an online PAC-Bayesian algorithm and draws links between PAC-Bayes and OL results. Sec. 4 details online PAC-Bayesian disintegrated procedures with reduced computational time and Sec. 5 gathers supporting experiments. We include reminders on OL and PAC-Bayes in Appendixes A.1 and C. Appendix B provide disucssion about our main result. All proofs are deferred to Appendix D.

## 2   An online PAC-Bayesian bound

We establish a novel PAC-Bayesian theorem (which in turn will be particularised in Sec. 3) which overcomes the classical limitation of data-independent prior and iid data. We call our main result an *online PAC-Bayesian bound* as it allows to consider a sequence of priors which may depend on the past and a sequence of posteriors that can dynamically evolve as well. Indeed, we follow the online learning paradigm which considers a continous stream of data that the algorithm has to process on the fly, adjusting its outputs at each time step w.r.t. the arrival of new data and the past. In the PAC-Bayesian framework, this paradigm translates as follows: from an initial (still data independent) prior $Q_1 = P$ and a data sample $S = (z_1, ..., z_m)$, we design a sequence of posterior $(Q_i)_{1 \leq i \leq m}$ where $Q_i = f(Q_1, ..., Q_{i-1}, z_i)$.

**Framework.** Consider a data space $\mathcal{Z}$ (which can be only inputs or pairs of inputs/outputs). We fix an integer $m > 0$ and our data sample $S \in \mathcal{Z}^m$ is drawn from an unknown distribution $\mu$. We do not make any assumption on $\mu$. We set a sequence of priors, starting with $P_1 = P$ a data-free distribution and $(P_i)_{i \geq 2}$ such that for each $i$, $P_i$ is $\mathcal{F}_{i-1}$ measurable where $(\mathcal{F}_i)_{i \geq 0}$ is an adapted filtration to $S$. For $P, Q \in \mathcal{M}_1(\mathcal{H})$, the notation $P \ll Q$ indicates that $Q$ is absolutely continuous wrt $P$ (i.e. $Q(A) = 0$ if $P(A) = 0$ for measurable $A \subset \mathcal{H}$). We also denote by $Q_i$ our sequence of candidate posteriors. There is no restriction on what $Q_i$ could be. In what follows we fix a filtration $(\mathcal{F}_i)_{i \geq 0}$ and we denote by KL the Kullback-Leibler divergence between two distributions.

We consider a predictor space $\mathcal{H}$ and a loss funtion $\ell : \mathcal{H} \times \mathcal{Z} \to \mathbb{R}^+$ bounded by a real constant $K > 0$. This loss defines the (potentially moving) learning objective. We denote by $\mathcal{M}_1(\mathcal{H})$ the set of all probability distributions on $\mathcal{H}$. We now introduce the notion of *stochastic kernel* [Rivasplata et al., 2020] which formalise properly data-dependent measures within the PAC-Bayes framework. First, for a fixed predictor space $\mathcal{H}$, we set $\Sigma_{\mathcal{H}}$ to be the considered $\sigma$-algebra on $\mathcal{H}$.

**Definition 2.1** (Stochastic kernels). *A stochastic kernel from $S = \mathcal{Z}^m$ to $\mathcal{H}$ is defined as a mapping $Q : \mathcal{Z}^m \times \Sigma_{\mathcal{H}} \to [0; 1]$ where*

- *For any $B \in \Sigma_{\mathcal{H}}$, the function $S = (z_1, ..., z_m) \mapsto Q(S, B)$ is measurable,*
- *For any $S \in \mathcal{Z}^m$, the function $B \mapsto Q(S, B)$ is a probability measure over $\mathcal{H}$.*

*We denote by $\mathtt{Stoch}(S, \mathcal{H})$ the set of all stochastic kernels from $S$ to $\mathcal{H}$ and for a fixed $S$, we set $Q_S := Q(S, .)$ the data-dependent prior associated to the sample $S$ through $Q$.*

From now, to refer to a distribution $Q_S$ depending on a dataset $S$, we introduce a stochastic kernel $Q(.,.)$ such that $Q_S = Q(S, .)$. Note that this notation is perfectly suited to the case when $Q_S$ is obtained from an algorithmic procedure $A$. In this case the stochastic kernel $Q$ of interest is the learning algorithm $A$. We use this notion to characterise our sequence of priors.

**Definition 2.2.** *We say that a sequence of stochastic kernels $(P_i)_{i=1..m}$ is an **online predictive sequence** if (i) for all $i \geq 1, S \in \mathcal{Z}^m, P_i(S, .)$ is $\mathcal{F}_{i-1}$ measurable and (ii) for all $i \geq 2, P_i(S, .) \gg P_{i-1}(S, .)$.*

Note that (ii) implies that for all $i, P_i(S, .) \gg P_1(S, .)$ with $P_1(S, .)$ a data-free measure (yet a classical prior in the PAC-Bayesian theory).

We can now state our main result.

**Theorem 2.3.** *For any distribution $\mu$ over $\mathcal{Z}^m$, any $\lambda > 0$ and any online predictive sequence (used as priors) $(P_i)$, for any sequence of stochastic kernels $(Q_i)$ we have with probability $1-\delta$ over the sample $S \sim \mu$, the following, holding for the data-dependent measures $Q_{i,S} := Q_i(S, .), P_{i,S} := P_i(S, .)$ :*

$$\sum_{i=1}^m \mathbb{E}_{h_i \sim Q_{i,S}} \left[ \mathbb{E}[\ell(h_i, z_i) \mid \mathcal{F}_{i-1}] \right] \leq \sum_{i=1}^m \mathbb{E}_{h_i \sim Q_{i,S}} \left[ \ell(h_i, z_i) \right] + \frac{\mathrm{KL}(Q_{i,S} \| P_{i,S})}{\lambda} + \frac{\lambda m K^2}{2} + \frac{\log(1/\delta)}{\lambda}.$$

**Remark 2.4.** *For the sake of clarity, we assimilate in what follows the stochastic kernels $Q_i, P_i$ to the data-dependent distributions $Q_i(S, .), P_i(S, .)$. Then, an online predictive sequence is also assimilated to a sequence of data-dependent distributions. Concretely this leads to the switch of notation $Q_{i,S} \to Q_i$ in Thm. 2.3. The reason of this switch is that, even though stochastic kernel is the right theoretical structure to state our main result, we consider in Secs. 3 and 4 practical algorithmic extensions which focus only on data-dependent distributions, hence the need to alleviate our notations.*

The proof is deferred to Appendix D.1. See Appendix B for context and discussions.

**A batch to online conversion.** First, we remark that our bound slightly exceeds the OL framework: indeed, it would require our posterior sequence to be an online predictive sequence as well, which is not the case here (for any $i$, the distribution $Q_{i,S}$ can depend on the whole dataset ). This is a consequence of our proof method (see Appendix D.1), which is classically denoted as a "batch to online" conversion (in opposition to the "online to batch" procedures as in Dekel and Singer, 2005). In other words, we exploited PAC-Bayesian tools designed for a fixed batch of data to obtain a dynamic result. This is why we refer to our bound as online as it allows to consider sequences of priors and posteriors that can dynamically evolve.

**Analysis of the different terms in the bound.** Our PAC-Bayesian bound formally differs in many points from the classical ones. On the left-hand side of the bound, the sum of the averaged expected loss conditioned to the past appears. Having such a sum of expectations instead of a single one is necessary to assess the quality of all our predictions. Indeed, because data may be dependent, one can not consider a single expectation as in the iid case. We also stress that taking an online predictive sequence as priors leads to control losses conditioned to the past, which differs from classical PAC-Bayes results designed to bound the expected loss. This term, while original in the PAC-Bayesian framework (to the best of our knowledge) recently appeared (in a modified form) in Wintenberger [2021, Prop 3]. See Appendix B.2 for further disucssions.

On the right hand-side of the bound, online counterparts of classical PAC-Bayes terms appear. At time $i$, the measure $Q_i$ (i.e. $Q_{i,S}$ according to Remark 2.4) has a tradeoff to achieve between an overfitted prediction of $z_i$ (the case $Q_i = \delta_{z_i}$ where $\delta$ is a Dirac measure) and a too weak impact of the new data with regards to our prior knowledge (the case $Q_i = P_i$). The quantity $\lambda > 0$ can be seen as a regulariser to adjust the relative impact of both terms.

**Influence of $\lambda$.** The quantity $\lambda$ also plays a crucial role on the bound as it is involved in an explicit tradeoff between the KL terms, the confidence term $\log(1/\delta)$ and the residual term $mK^2/2$. This idea of seeing $\lambda$ as a trading parameter is not new [Thiemann et al., 2017, Germain et al., 2016]. However, the results from Thiemann et al. [2017] stand w.p. $1-\delta$ for any $\lambda$ while ours and the ones from Germain et al. [2016] hold for any $\lambda$ w.p. $1-\delta$ which is weaker and implies to discretise $\mathbb{R}^+$ onto a grid to estimate the optimal $\lambda$.

We now move on to the design of online PAC-Bayesian algorithms.

# 3 An online PAC-Bayesian procedure

OL algorithms (we refer to Hazan, 2016 an introduction to the field) are producing sequences of predictors by progressively updating the considered predictor (see Appendix A.1 for an example). Recall that, in the OL framework, an algorithm outputs at time $i$ a predictor which is $\mathcal{F}_{i-1}$-measurable. Here, our goal is to design an online procedure derived from Thm. 2.3 which outputs an online predictive sequence (which is assimilated, according to Remark 2.4, to a sequence of distributions).

**Online PAC-Bayesian (OPB) training bound.** We state a corollary of our main result which paves the way to an online algorithm. This constructive procedure motivates the name *Online PAC-Bayesian training bound* (OPBTRAIN in short).

**Corollary 3.1** (OPBTRAIN). *For any distribution $\mu$ over $\mathcal{Z}^m$, any $\lambda > 0$ and any online predictive sequences $\hat{Q}, P$, the following holds with probability $1 - \delta$ over the sample $S \sim \mu$ :*

$$\sum_{i=1}^{m} \mathbb{E}_{h_i \sim \hat{Q}_{i+1}} \left[ \mathbb{E}[\ell(h_i, z_i) \mid \mathcal{F}_{i-1}] \right] \leq \sum_{i=1}^{m} \mathbb{E}_{h_i \sim \hat{Q}_{i+1}} \left[ \ell(h_i, z_i) \right] + \frac{\mathrm{KL}(\hat{Q}_{i+1} \| P_i)}{\lambda} + \frac{\lambda m K^2}{2} + \frac{\log(1/\delta)}{\lambda}.$$

Here, $\lambda$ is seen as a scale parameter as precised below. The proof consists in applying Thm. 2.3 with for all $i$, $Q_i = \hat{Q}_{i+1}$ and $P_i$. Note that in this case, our posterior sequence is an online predictive sequence in order to fit with the OL framework.

Corollary 3.1 suggests to design $\hat{Q}$ as follows, assuming we have drawn a dataset $S = \{z_1, ..., z_m\}$, fixed a scale parameter $\lambda > 0$ and an online predictive sequence $P_i$:

$$\hat{Q}_1 = P_1, \quad \forall i \geq 1 \ \hat{Q}_{i+1} = \underset{Q \in \mathcal{M}_1(\mathcal{H})}{\mathrm{argmin}} \ \mathbb{E}_{h_i \sim Q} \left[ \ell(h_i, z_i) \right] + \frac{\mathrm{KL}(Q \| P_i)}{\lambda} \tag{1}$$

which leads to the explicit formulation

$$\frac{d\hat{Q}_{i+1}}{dP_i}(h) = \frac{\exp\left(-\lambda \ell(h, z_i)\right)}{\mathbb{E}_{h \sim P_i} \left[ \exp\left(-\lambda \ell(h, z_i)\right) \right]}. \tag{2}$$

Thus, the formulation of Eq. (2), which has been highlighted by Catoni [2003, Sec. 5.1] shows that our online procedure produces Gibbs posteriors. So, PAC-Bayesian theory provides sound justification for the somewhat intuitive online procedure in Eq. (1): at time $i$, we adjust our new measure $\hat{Q}_{i+1}$ by optimising a tradeoff between the impact of the newly arrived data $z_i$ and the one of prior knowledge $\hat{Q}_i$.

Notice that $\hat{Q}$ is an online predictive sequence: $\hat{Q}_i$ is $\mathcal{F}_{i-1}$-measurable for all $i$ as it depends only on $\hat{Q}_{i-1}$ and $z_{i-1}$. Furthermore, one has $\hat{Q}_i \gg \hat{Q}_{i-1}$ for all $i$ as $\hat{Q}_i$ is defined as an argmin and the KL term is finite if and only it is absolutely continuous w.r.t. $\hat{Q}_{i-1}$.

**Remark 3.2.** *In Corollary 3.1, while the right hand-side is the reason we considered Eq. (1), the left hand side still needs to be analysed. It expresses how the posterior $\hat{Q}_{i+1}$ (designed from $\hat{Q}_i, z_i$) generalises well on average to any new draw of $z_i$. More precisely, this term measures how much the training of $\hat{Q}_{i+1}$ is overfitting on $z_i$. A low value of it ensures our online predictive sequence, which is obtained from a single dataset, is robust to the randomness of $S$, hence the interest of optimising the right hand side of the bound. This is a supplementary reason we refer to Corollary 3.1 as an* OPBTRAIN *bound as it provide robustness guarantees for our training.*

**Online PAC-Bayesian (OPB) test bound.** However, Corollary 3.1 does not say if $\hat{Q}_{i+1}$ will produce good predictors to minimise $\ell(., z_{i+1})$, which is the objective of $\hat{Q}_{i+1}$ in the OL framework (we only have access to the past to predict the future). We then need to provide an *Online PAC-Bayesian (OPB) test bound* (OPBTEST bound) to quantify our prediction's accuracy. We now derive an OPBTEST bound from Thm. 2.3.

**Corollary 3.3** (OPBTEST). *. For any distribution $\mu$ over $\mathcal{Z}^m$, any $\lambda > 0$, and any online predictive sequence $(\hat{Q}_i)$, the following holds with probability $1 - \delta$ over the sample $S \sim \mu$:*

$$\sum_{i=1}^{m} \mathbb{E}_{h_i \sim \hat{Q}_i} \left[ \mathbb{E}[\ell(h_i, z_i) \mid \mathcal{F}_{i-1}] \right] \leq \sum_{i=1}^{m} \mathbb{E}_{h_i \sim \hat{Q}_i} \left[ \ell(h_i, z_i) \right] + \frac{\lambda m K^2}{2} + \frac{\log(1/\delta)}{\lambda}.$$

*Optimising in $\lambda$ gives $\lambda = \sqrt{2\log(1/\delta)/mK^2}$ and ensure that:*

$$\sum_{i=1}^{m} \mathbb{E}_{h_i \sim \hat{Q}_i} \left[ \mathbb{E}[\ell(h_i, z_i) \mid \mathcal{F}_{i-1}] \right] \leq \sum_{i=1}^{m} \mathbb{E}_{h_i \sim \hat{Q}_i} \left[ \ell(h_i, z_i) \right] + O\left( \sqrt{\log(1/\delta)K^2 m} \right).$$

The proof consists in applying Thm. 2.3 with for all $i$, $Q_i = \hat{Q}_i = P_i$.

Corollary 3.3 quantifies how efficient will our predictions be. Indeed, the left hand side of this bound relates for all $i$, how good $\hat{Q}_i$ is to predict $z_i$ (on average) which is what $\hat{Q}_i$ is designed for. Note that here, the involved $\lambda$ can differ from the scale parameter of Eq. (1), it is now a way to compensate for the tradeoff between the two last terms of the bound. The strength of this bound is that since $\hat{Q}$ is an online predictive sequence, the Kullback-Leibler terms vanished, leaving terms depending only on hyperparameters.

**Links with previous approaches**

We now present a specific case of Corollary 3.1 where we choose as priors the online predictive sequence $\hat{Q}$ (*i.e.* in Thm. 2.3, we choose $Q_i = \hat{Q}_{i+1}, P_i = \hat{Q}_i$). The reason we focus on this specific case is that it enables to build strong links between PAC-Bayes and OL.

We then adapt our OPBTRAIN bound (Corollary 3.1). The online procedure becomes:

$$\hat{Q}_1 = P, \quad \forall i \geq 1 \ \hat{Q}_{i+1} = \operatorname{argmin}_Q \mathbb{E}_{h_i \sim Q} \left[ \ell(h_i, z_i) \right] + \frac{\mathrm{KL}(Q \| \hat{Q}_i)}{\lambda}, \tag{3}$$

which leads to the explicit formulation

$$\frac{d\hat{Q}_{i+1}}{d\hat{Q}_i}(h) = \frac{\exp\left(-\lambda \ell(h, z_i)\right)}{\mathbb{E}_{h \sim \hat{Q}_i}\left[\exp\left(-\lambda \ell(h, z_i)\right)\right]}.$$

**Links with classical PAC-Bayesian bounds.** We denote that the optimal predictor in this case is such that at any time $i$, $d\hat{Q}_{i+1}(h) \propto \exp(-\lambda \ell(h, z_i))d\hat{Q}_i(h)$ hence $d\hat{Q}_{m+1}(h) \propto \exp\left(-\lambda \sum_{i=1}^{m} \ell(h, z_i)\right) d\hat{Q}_1(h)$. One recognises, up to a multiplicative constant, the optimised predictor of Catoni [2007, Th 1.2.6] which solves $\operatorname{argmin}_Q \mathbb{E}_{h \sim Q} \left[ \frac{1}{m} \sum_{i=1}^{m} \ell(h, z_i) \right] + \frac{\mathrm{KL}(Q \| \hat{Q}_1)}{\lambda}$, thus one sees that in this case, the output of our online procedure after $m$ steps coincides with Catoni's output. This shows consistency of our general procedure which recovers classical result within an online framework: when too many data are available, treating data sequentially until time $m$ leads to the same Gibbs posterior than if we were treating the whole dataset as a batch.

**Analogy with Online Gradient Descent (OGD).** We propose an analogy between the procedure Eq. (3) and the celebrated OGD algorithm (see Appendix A.1 for a recap). First we remark that our minimisation problem is equivalent to $\operatorname{argmin}_Q \lambda \mathbb{E}_{h_i \sim Q} \left[ \ell(h_i, z_i) \right] + \mathrm{KL}(Q \| \hat{Q}_i)$. Then we assume that for any $i, \hat{Q}_i = \mathcal{N}(\hat{m}_i, I_d)$ with $\hat{m}_i \in \mathbb{R}^d$ and we set $\mathcal{L}_i(\hat{m}_i) = \mathbb{E}_{h_i \sim \hat{Q}_i} \left[ \ell(h_i, z_i) \right]$. The minimisation problem becomes: $\operatorname{argmin}_{\hat{m}} \lambda \mathcal{L}_i(\hat{m}) + \frac{1}{2} \| \hat{m} - \hat{m}_i \|^2$. And so using the first order Taylor expansion, we use the approximation $\mathcal{L}_i(\hat{m}) \approx \mathcal{L}_i(\hat{m}_i) + \langle \hat{m} - \hat{m}_i, \nabla \mathcal{L}_i(\hat{m}_i) \rangle$ which finally transform our argmin into the following optimisation process: $\hat{m}_{i+1} = \hat{m}_i - \lambda \nabla \mathcal{L}_i(\hat{m}_i)$ which is exactly OGD on the loss sequence $\mathcal{L}_i$. We draw an analogy between the scale parameter $\lambda$ and the step size $\eta$ in OGD. the KL term translates the influence of the previous point and the expected loss gives the gradient. This analogy has been already exploited in Shalev-Shwartz [2012] where they approximated $\mathbb{E}_{h_i \sim q_\mu}[\ell(h_i, z_i)] := \bar{L}_i(\mu) \approx \mu^T \nabla \bar{L}_i(\mu_i)$ where $\mu$ is their considered online predictive sequence.

Finally, we remark that the optimum rate in Corollary 3.3 is a $O(\sqrt{m})$ which is comparable to the best rate of Shalev-Shwartz [2012, Eq (2.5)] (see proposition A.2).

**Comparison with previous work.** We acknowledge that the procedure of Eq. (3) already appeared in literature. Li et al. [2018, Alg. 1] propose a Gibbs procedure somewhat similar to ours, the main difference being the addition of a surrogate of the true loss at each time step. Within the OL literature, the idea of updating measures online has been recently studied for instance in Chérief-Abdellatif et al. [2019]. More precisely, our procedure is similar to their Streaming Variational Bayes (SVB) algorithm. A slight difference is that they approximated the expected loss similarly to Shalev-Shwartz [2012]. The guarantees Chérief-Abdellatif et al. [2019] provided for SVB hold for Gaussian priors and comes at the cost of additional constraints that do not allow to consider any aggregation strategies contrary to what Corollary 3.1 propose. Their bounds are deterministic and are using tools and assumptions from convex optimisation (such that convex expected losses) while ours are probabilistic and are using measure theory tools which allow to relax these assumptions.

**Strength of our result.** We emphasize two points. First, to the best of our knowledge, Corollary 3.1 is the first bound which theoretically suggests Eq. (3) as a learning algorithm. Second, we stress that Eq. (3) is a particular case of Corollary 3.1 and our result can lead to other fruitful routes. For instance, we consider the idea of adding noise to our measures at each time step to avoid overfitting (this idea has been used *e.g.* in Neelakantan et al., 2015 in the context of deep neural networks): if our online predicitve sequence $(\hat{Q}_i)$ can be defined through a sequence of parameter vectors $\hat{\mu}$, then we can define $P_i$ by adding a small noise on $\hat{\mu}_i$ and thus giving more freedom through stochasticity.

Thus, we see that our procedure led us to the use of the Gibbs posteriors of Catoni. However, in practice, Gaussian distributions are preferred [*e.g.* Dziugaite and Roy, 2017, Rivasplata et al., 2019, Perez-Ortiz et al., 2021b,a, Pérez-Ortiz et al., 2021]). That is why we focus next on new online PAC-Bayesian algorithms involving Gaussian distributions.

## 4 Disintegrated online algorithms for Gaussian distributions.

We dig deeper in the field of disintegrated PAC-Bayesian bounds, originally explored by Catoni [2007], Blanchard and Fleuret [2007], further studied by Alquier and Biau [2013], Guedj and Alquier [2013] and recently developed by Rivasplata et al. [2020], Viallard et al. [2021] (see Appendix C for a short presentation of the bound we adapted and used). The strength of the disintegrated approach is that we have directly guarantees on the random draw of a single predictor, which avoids to consider expectations over the predictor space. This fact is particularly significant in our work as the procedure precised in Eq. (2), require the estimation of an exponential moment to be efficient, which may be costful. We then show that disintegrated PAC-Bayesian bounds can be adapted to the OL framework, and that they have the potential to generate proper online algorithms with weak computational cost and sound efficiency guarantees.

**Online PAC-Bayesian disintegrated (OPBD) training bounds.** We present a general form for *online PAC-Bayes disintegrated (OPBD) training bounds*. The terminology comes from the way we craft those bounds: from PAC-Bayesian disintegrated bounds we use the same tools as in Thm. 2.3 to create the first online PAC-Bayesian disintegrated bounds. OPBD training bounds have the following form.

For any online predictive sequences $\hat{Q}, P$, any $\lambda > 0$ w.p. $1 - \delta$ over $S \sim \mu$ and $(h_1, ..., h_m) \sim \hat{Q}_2 \otimes ... \otimes \hat{Q}_{m+1}$:

$$\sum_{i=1}^m \mathbb{E}[\ell(h_i, z_i) \mid \mathcal{F}_{i-1}] \leq \sum_{i=1}^m \ell(h_i, z_i) + \Psi(h_i, \hat{Q}_{i+1}, P_i) + \Phi(m), \tag{4}$$

with $\Psi, \Phi$ being real-valued functions. $\Psi$ controls the global behaviour of $Q_{i+1}$ w.r.t. the $\mathcal{F}_{i-1}$-measurable prior $P_i$. If one has no dependency on $h_i$ this behaviour is global, otherwise it is local. Note that those functions may depend on $\lambda, \delta$. However, since they are fixed parameters, we do not make these dependencies explicit. Similarly to Corollary 3.1, this kind of bound allows to derive a learning algorithm (cf Algorithm 1) which outputs an online predicitve sequence $\hat{Q}$. Finally we draw $(h_1, ..., h_m) \sim \hat{Q}_2 \otimes ... \otimes \hat{Q}_{m+1}$ (and not $\hat{Q}_1 \otimes ... \otimes \hat{Q}_m$) since an OPBD bound is designed to justify theoretically an OPBD procedure in the same way Corollary 3.1 allowed to justify Eq. (1).

**Why focus on Gaussian measures?** The reason is that a Gaussian variable $h \sim \mathcal{N}(w, \sigma^2 \mathbf{I}_d)$ can be written as $h = w + \varepsilon$ with $\varepsilon \sim \mathcal{N}(0, \sigma^2 \mathbf{I}_d)$, and this expression totally defines $h$ ($\mathbf{I}_d$ being the identity matrix).

**A general OPBD algorithm for Gaussian measure with fixed variance** We use an idea presented in Viallard et al. [2021] which restrict the measure set to Gaussian on $\mathbb{R}^d$ **with known and fixed covariance matrix** $\sigma^2 \mathbf{I}_d$. Then we present in Algorithm 1 a general algorithm (derived from an OPBD training bound) for Gaussian measures with fixed variance which outputs a sequence of gaussian $\hat{Q}_i = \mathcal{N}(\hat{w}_i, \sigma^2 \mathbf{I}_d)$ from a prior sequence $P_i = \mathcal{N}(w_i^0, \sigma^2 \mathbf{I}_d)$ where for each $i$, $w_i^0$ is $\mathcal{F}_{i-1}$-measurable. Because the variance is fixed, the distribution is uniquely defined by its mean, thus we identify $\hat{Q}_i$ and $\hat{w}_i$, $P_i$ and $w_i^0$.

---

**Algorithm 1:** A general OPBD algorithm for Gaussian measures with fixed variance.

**Parameters** : Time m, scale parameter $\lambda$
**Initialisation** : Variance $\sigma^2$, Initial mean $\hat{w}_1 \in \mathbb{R}^d$, epoch $m$
1 **for** *each iteration $i$ in $1..m$* **do**
2 $\quad$ Observe $z_i, w_i^0$ and draw $\varepsilon_i \sim \mathcal{N}(0, \sigma^2 \mathbf{I}_d)$
3 $\quad$ Update:
$$\hat{w}_{i+1} := \mathrm{argmin}_{w \in \mathbb{R}^d}\, \ell(w + \varepsilon_i, z_i) + \Psi(w + \varepsilon_i, w, w_i^0)$$
4 **end**
5 **Return** $(\hat{w}_i)_{i=1..m+1}$

---

At each time $i$, Algorithm 1 requires the draw of $\varepsilon_i \sim \mathcal{N}(0, \sigma^2 \mathbf{I}_d)$. Doing so, we generated the randomness for our $h_i$ (because our bound holds for a single draw of $(h_1, .., h_m) \sim \hat{Q}_2 \otimes ... \otimes \hat{Q}_{m+1}$), we then write $h_i = w + \varepsilon_i$ and we optimise w.r.t. $\Psi$ to find $\hat{w}_{i+1}$.

**Bounds of interest.** We present two possible choices of pairs $(\Psi, \Phi)$ derived from the disintegrated results presented in Appendix C. Doing so, we explicit two ready-to-use declinations of Algorithm 1.

**Corollary 4.1.** *For any distribution $\mu$ over $\mathcal{Z}^m$, any online predictive sequences of Gaussian measures with fixed variance $\hat{Q}_i = \mathcal{N}(\hat{w}_i, \sigma^2 \mathbf{I}_d)$ and $P_i = \mathcal{N}(w_i^0, \sigma^2 \mathbf{I}_d)$, any $\lambda > 0$, w.p. $1 - \delta$ over $S \sim \mu$ and $(h_i = \hat{w}_{i+1} + \varepsilon_i)_{i=1..m} \sim \hat{Q}_2 \otimes ... \otimes \hat{Q}_{m+1}$, the bound of Eq. (4) holds for the two following pairs $\Psi, \Phi$:*

$$\Psi_1(h_i, \hat{w}_{i+1}, w_i^0) = \frac{1}{\lambda} \left( \frac{||\hat{w}_{i+1} + \varepsilon_i - w_i^0||^2 - ||\varepsilon||^2}{2\sigma^2} \right) \quad \Phi_1(m) = \frac{\lambda m K^2}{2} + \frac{\log(1/\delta)}{\lambda}, \quad (5)$$

$$\Psi_2(h_i, \hat{w}_{i+1}, w_i^0)) = \frac{1}{\lambda} \frac{||\hat{w}_{i+1} - w_i^0||^2}{2\sigma^2} \quad \Psi_2(m) = \lambda m K^2 + \frac{3\log(1/\delta)}{2\lambda}. \quad (6)$$

*Where the notation $1, 2$ denote whether the functions have been derived from adapted theorems of Rivasplata et al., 2020, Viallard et al., 2021 recalled in Appendix C We then can use algorithm 1 with Eq. (5), Eq. (6).*

Proof is deferred to Appendix D.2. Note that in Corollary 4.1, we identified $\hat{Q}_i$ to $\hat{w}_i$ and for the last formula, $\Psi$ has no dependency on $h_i$.

**Comparison with Eq. (1).** The main difference with Eq. (1) provided by the disintegrated framework is that the optimisation route does not include an expected term within the optimisation objective. The main advantage is a weaker computational cost when we restrict to Gaussian distributions. The main weakness is a lack of stability as our algorithm now depends at time $i$ on $\ell(h + \varepsilon_i, z_i)$ so on $\varepsilon_i$ directly. We denote that Eq. (5) is less stable than Eq. (6) as it involves another dependency on $\varepsilon_i$ through $\Psi$. The reason is that Rivasplata et al. [2020] proposed a bound involving a disintegrated KL divergence while Viallard et al. [2021] proposed a result involving a Rényi divergence avoiding a dependency on $\varepsilon_i$. We refer to Appendix C for a detailed statement of those properties.

**Comparison with van der Hoeven et al. [2018].** Theorem 3 of van der Hoeven et al. [2018] recovers OGD from the exponential weights algorithm by taking a sequence of moving distributions being Gaussians with fixed variance which is exactly what we consider here. From these, they retrieve the classical OGD algorithm as well as its classical convergence rate. Let us compare our results with theirs.

First, if we fix a single step $\eta$ in their bound and assume two traditional assumptions for OGD (a finite diameter $D$ of the convex set and an uniform bound $G$ on the loss gradients), we recover

for the OGD (greedy GD in van der Hoeven et al., 2018) a rate of $\frac{D^2}{2\sigma^2\eta} + \frac{\eta\sigma^2 TG^2}{2}$. This is, up to constants and notation changes, exactly our $\Psi_i$ ($i \in \{1, 2\}$). Also, we notice a difference in the way to use Gaussian distributions: Theorem 3 of van der Hoeven et al. [2018] is based on their Lemma 1 which provides guarantees for the expected regret. This is a clear incentive to consider as predictors the mean of the sucessive Gaussians of interest. On the contrary, Corollary 4.1 involves a supplementary level of randomness by considering predictors $h_i$ drawn from our Gaussians. This additional randomness appears in our optimisation process (algorithm 1). Finally, notice that van der Hoeven et al. [2018] based their whole work on the use of a KL divergence while Corollary 4.1 not only exploit a disintegrated KL ($\Psi_1$) but also a Rényi $\alpha$-divergence ($\Psi_2$). Note that we propose a result only for $\alpha = 2$ for the sake of space constraints but any other value of $\alpha$ leads to another optimisation objective to explore.

**OPBD test bounds.** Similarly to what we did in Sec. 3, we also provide *OPBD test bounds* to provide efficiency guarantees for online predicitve sequences (e.g. the output of algorithm 1). Our proposed bounds have the following general form.

For any online predictive sequence $\hat{Q}$, any $\lambda > 0$ w.p. $1 - \delta$ over $S$ and $(h_1, ..., h_m) \sim \hat{Q}_1 \otimes ... \otimes \hat{Q}_m$:

$$\sum_{i=1}^{m} \mathbb{E}[\ell(h_i, z_i) \mid \mathcal{F}_{i-1}] \leq \sum_{i=1}^{m} \ell(h_i, z_i) + \Phi(m), \tag{7}$$

with $\Phi$ being a real-valued function(possibly dependent on $\lambda, \delta$ though it is not explicited here).

Note that our predictors $(h_1, ..., h_m)$ are now drawn from $\hat{Q}_1 \otimes ... \otimes \hat{Q}_m$. Thus, the left-hand side of the bound considers a $h_i$ drawn from an $\mathcal{F}_{i-1}$-measurable distribution evaluated on $\ell(., z_i)$: this is effectively a measure of the prediction performance.

We now state a corollary which gives disintegrated guarantees for any online predicitve sequence.

**Corollary 4.2.** *For any distribution $\mu$ over $\mathcal{Z}^m$, any $\lambda > 0$, and any online predictive sequence $(\hat{Q}_i)$, the following holds with probability $1 - \delta$ over the sample $S \sim \mu$ and the predictors $(h_1, ..., h_m) \sim \hat{Q}_1 \otimes ... \otimes \hat{Q}_m$, the bound of Eq. (7) holds with :*

$$\Phi_1(m) = \frac{\lambda m K^2}{2} + \frac{\log(1/\delta)}{\lambda}, \quad \Phi_2(m) = 2\lambda m K^2 + \frac{\log(1/\delta)}{\lambda}.$$

*Where the notation $1, 2$ denote whether the functions have been derived from adapted theorems of Rivasplata et al., 2020, Viallard et al., 2021 recalled in Appendix C. The optimised $\lambda$ gives in both cases a $O(\sqrt{m \log(1/\delta)})$.*

Proof is deferred to Appendix D.2.

## 5   Experiments

We adapt the experimental framework introduced in Chérief-Abdellatif et al. [2019, Sec.5] to our algorithms (anonymised code available here). We conduct experiments on several real-life datasets, in classification and linear regression. Our objective is twofold: check the convergence of our learning methods and compare their efficiencies with classical algorithms. We first introduce our experimental setup.

**Algorithms.** We consider four online methods of interest: the OPB algorithm of Eq. (3) which update through time a Gibbs posterior. We instantiate it with two different priors $\hat{Q}_1$: a Gaussian distribution and a Laplace one. We also implement Algorithm 1 with the functions $\Psi_1, \Psi_2$ from Corollary 4.1. To assess efficiency, we implement the classical OGD (as described in Alg. 1 of Zinkevich, 2003) and the SVB method of Chérief-Abdellatif et al. [2019].

**Binary Classification.** At each round $i$ the learner receives a data point $x_i \in \mathbb{R}^d$ and predicts its label $y_i \in \{-1, +1\}$ using $\langle x_i, h_i \rangle$, with $h_i = \mathbb{E}_{h \sim \hat{Q}_i}[h]$ for OPB methods or $h_i$ being drawn under $\hat{Q}_i$ for OPBD methods. The adversary reveals the true value $y_i$, then the learner suffers the loss $\ell(h_i, z_i) = (1 - y_i h_i^T x_i)_+$ with $z_i = (x_i, y_i)$ and $a_+ = a$ if $a > 0$ and $a_+ = 0$ otherwise. This loss is unbounded but can be thresholded.

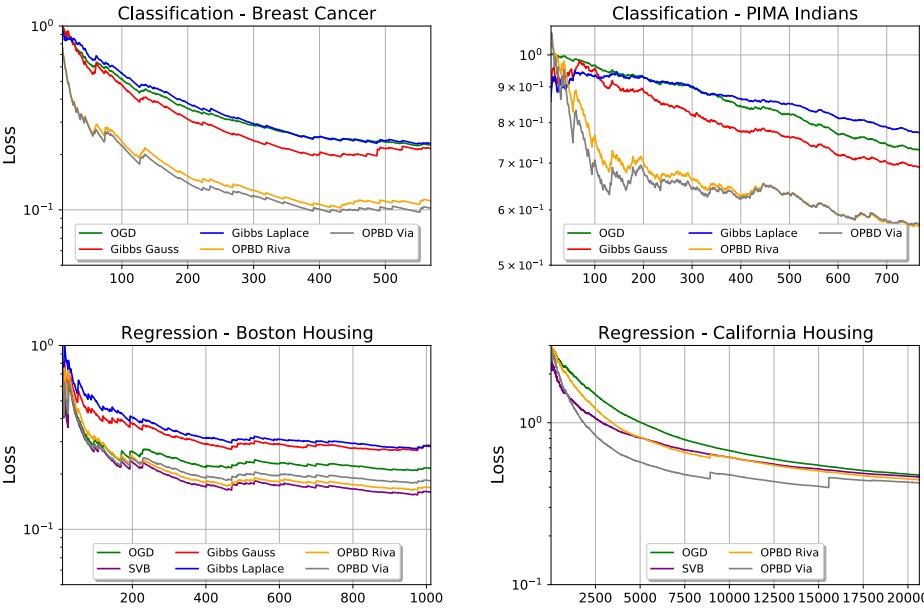

Figure 1: Averaged cumulative losses for all four considered datasets. 'Gibbs Gauss' denotes OPB with Gaussian Prior, 'Gibbs Laplace' denotes OPB with Laplace prior. 'OPBD Riva' denotes OPBD with $\Psi_1$, 'OPBD Via' denotes OPBD with $\Psi_2$.

**Linear Regression.** At each round $i$, the learner receives a set of features $x_i \in \mathbb{R}^d$ and predicts $y_i \in \mathbb{R}$ using $\langle x_i, h_i \rangle$ with $h_i = \mathbb{E}_{h \sim \hat{Q}_i}[h]$ for SVB and OPB methods or $h_i$ being drawn under $\hat{Q}_i$ for OPBD methods. Then the adversary reveals the true value $y_t$ and the learner suffers the loss $\ell(h_i, z_i) = \left(y_i - h_i^T x_i\right)^2$ with $z_i = (x_i, y_i)$. This loss is unbounded but can be thresholded.

**Datasets.** We consider four real world dataset: two for classification (Breast Cancer and Pima Indians), and two for regression (Boston Housing and California Housing). All datasets except the Pima Indians have been directly extracted from `sklearn` [Pedregosa et al., 2011]. Breast Cancer dataset [Street et al., 1993] is available here and comes from the UCI ML repository as well as the Boston Housing dataset [Belsley et al., 2005] which can be obtained here. California Housing dataset [Pace and Barry, 1997] comes from the StatLib repository and is available here. Finally, Pima Indians dataset [Smith et al., 1988] has been recovered from this Kaggle repository. Note that we randomly permuted the observations to avoid to learn irrelevant human ordering of data (such that date or label).

**Parameter settings.** We ran our experiments on a 2021 MacBookPro with an M1 chip and 16 Gb RAM. For OGD, the initialisation point is $\mathbf{0}_{\mathbb{R}^d}$ and the values of the learning rates are set to $\eta = 1/\sqrt{m}$. For SVB, mean is initialised to $\mathbf{0}_{\mathbb{R}^d}$ and covariance matrix to $\mathrm{Diag}(1)$. Step at time $i$ is $\eta_i = 0.1/\sqrt{i}$. For both of the OPB algorithms with Gibbs posterior, we chose $\lambda = 1/m$. As priors, we took respectively a centered Gaussian vector with the covariance matrix $\mathrm{Diag}(\sigma^2)$ ($\sigma = 1.5$) and an iid vector following the standard Laplace distribution. For the OPBD algorithm with $\Psi_1$, we chose $\lambda = 10^{-4}/m$, the initial mean is $\mathbf{0}_{\mathbb{R}^d}$ and our fixed covariance matrix is $\mathrm{Diag}(\sigma^2)$ with $\sigma = 3.10^{-3}$. For the OPBD algorithm with $\Psi_1$, we chose $\lambda = 2.10^{-3}/m$, the initial mean is $\mathbf{0}_{\mathbb{R}^d}$ and our covariance matrix is $\mathrm{Diag}(\sigma^2)$ with $\sigma = 10^{-2}$. The reason of those higher scale parameters and variance is that $\Psi$ from Rivasplata et al. [2020] is more stochastic (yet unstable) than the one Viallard et al. [2021].

**Experimental results.** For each dataset, we plot the evolution of the average cumulative loss $\sum_{i=1}^t \ell(h_i, z_i)/t$ as a function of the step $t = 1, \ldots, m$, where $m$ is the dataset size and $h_i$ is the decision made by the learner $h_i$ at step $i$. The results are gathered in Fig. 1

**Empirical findings.** OPB with Gaussian prior ('Gibbs Gauss') outperforms OGD on all datasets except California Housing (on which this method is not implemented ) while OPB with Laplace prior ('Gibbs Laplace') always fail w.r.t. OGD. OPB methods fail to compete with SVB on the Boston

Housing dataset. OPBD methods compete with SVB on regression problems and clearly outperforms OGD on classification tasks. OPBD with $\Psi_2$ (labeled as 'OPBD Via' in Fig. 1) performs better on the California Housing dataset while OPBD with $\Psi_1$ (labeled as 'OPBD Riva') is more efficient on the Boston Housing dataset. Both methods performs roughly equivalently on classification tasks. This brief experimental validation shows the consistency of all our online procedures as we observe a visible decrease of the cumulative losses through time. It particularly shows that OPBD procedures improve on OGD on these dataset. We refer to Appendix E for additional table gathering the error bars of our OPBD methods.

**Why do we perform better than OGD?** As stated in Sec. 4, OGD can be recovered as a Gaussian approximation of the exponential weights algorithm (EWA). Thus, a legitimate question is why do we perform better than OGD as our OPBD methods are also based on a Gaussian surrogate of EWA? van der Hoeven et al. [2018] only used Gaussians distributions with fixed variance as a technical tool when the considered predictors are the Gaussian means. In our work, we exploited a richer characteristic of our distributions in the sense our predictors are points sampled from our Gaussians and not only the means. This also has consequences in our learning algorithm as at time $i$ of our algorithm 1, our optimisation step involves a noise $\varepsilon_i \sim \mathcal{N}(0, \sigma^2 \mathbf{I})$. Thus, we believe that OPBD methods should perform at least as well as OGD. We write 'at least' as we think that the higher flexibility due to this additional level of randomness might result in slightly better empirical performances, as seen on the few datasets in Fig. 1.

## 6 Conclusion

We establish links between Online Learning and PAC-Bayes. We show that PAC-bayesian bounds are useful to derive new OL algorithms. We also prove sound theoretical guarantees for such algorithms. We emphasise that all of our results stand for any general bounded loss, especially no convexity assumption is needed. Having no convexity assumption on the loss paves the way to exciting future practical studies, starting with *Spiking Neural Network* which is investigated in an online fashion (see Lobo et al., 2020 for a recent survey). A follow-up question on the theoretical part is whether we can relax the bounded loss assumption: we leave this for future work.

## Acknowledgements

We warmly thank reviewers and the Area Chair who provided insigthful comments and suggestions which greatly helped us improve our manuscript. B.G. acknowledges partial support by the U.S. Army Research Laboratory and the U.S. Army Research Office, and by the U.K. Ministry of Defence and the U.K. Engineering and Physical Sciences Research Council (EPSRC) under grant number EP/R013616/1; B.G. also acknowledges partial support from the French National Agency for Research, grants ANR-18-CE40-0016-01 and ANR-18-CE23-0015-02.

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
