# A Background

## A.1 Reminder on Online Gradient Descent

For the sake of completeness we re-introduce the projected Online Gradient Descent (OGD) on a convex set $\mathcal{K}$. This is a first example of online learning philosophy. It may be the algorithm that applies to the most general setting of online convex optimization. This algorithm, which is based on standard gradient descent from offline optimization, was introduced in its online form by Zinkevich [2003]. In each iteration, the algorithm takes a step from the previous point in the direction of the gradient of the previous cost. This step may result in a point outside of the underlying convex set. In such cases, the algorithm projects the point back to the convex set, i.e. finds its closest point in the convex set. We precise this algorithm works with the assumptions of a convex set $\mathcal{K}$ bounded in diameter by $D$ and of bounded gradients (by a certain $G$). We also assume here to have a dataset $S = (z_t)_{t=1..T}$ and to be coherent with the online learning philosophy, we assume that for each $t > 0$, we possess a loss function $\ell_t$ depending on the points $(z_1, ...z_t)$. We present OGD in algorithm 2

---

**Algorithm 2:** Projected OGD onto a convex $\mathcal{K}$ with fixed step $\eta$.

**Parameters** : Epoch T, step-size $(\eta)$
**Initialisation** : Convex set $\mathcal{K}$, Initial point $\theta_0 \in \mathcal{K}$, T, step sizes $(\eta_t)_t$
1 **for** *each iteration $t$ in $1..T$* **do**
2 $\quad$ Compute $f'(\theta_n)$
3 $\quad$ Play (observe) $\theta_t$ and compute the cost $f_t(\theta_t)$ Update and project

$$\zeta_t = \theta_{t-1} - \eta \nabla \ell_t(\theta_{t-1})$$

$$\theta_t = \Pi_{\mathcal{K}}(\zeta_t)$$

4 **end**
5 **Return** $\theta_T$

---

One now defines the notion of regret which is the classical quantity to evaluate the performance of an online algorithm.

**Definition A.1.** *One defines the* regret *of a decision sequence $(\theta_t)$ at time $T$ w.r.t. a point $\theta$ as:*

$$Regret_T(\theta) := \sum_{t=1}^{T} \ell_t(\theta_t) - \sum_{t=1}^{T} \ell_t(\theta)$$

Now we state a regret bound which can be found in [Shalev-Shwartz, 2012, Eq 2.5] although we slightly modified the result, which uses additional hypotheses from Hazan [2016].

**Proposition A.2.** *Assume that $\mathcal{K}$ has a fixed diameter $D$ and that the gradients of any point is bounded by $G$. Then for any $\theta \in \mathcal{K}$, the regret of projected OGD with fixed step $\eta$ satisfies:*

$$Regret_T(\theta) \leq \frac{D^2}{2\eta} + \eta T G^2$$

## A.2 About PAC-Bayes learning

We provide here more information about PAC-Bayes learning. We propose a local framework; then expose what APC-Bayes lerning ais to do and finally state two celebrated PAC-Bayes theorems.

**An usual framework** We first state the following framework:

- $\mathcal{H}$ is a space of considered predictors
- $\mathcal{Z}$ is a data space. $z$ can be an unlabeled data $x$ or a couple $(x, y)$ of a point with its label. We assume that $\mu$ is a distribution over $\mathcal{Z}$ which rules the distribution of our data.
- $\ell : \mathcal{H} \times \mathcal{Z} \to \mathbb{R}^+$ is a loss function i.e. the learning objective we want to minimise.

- $S = (z_1, ... z_m)$ an iid dataset following $\mu$.
- The generalisation risk for $h \in \mathbb{H}$: $R(h) = \mathbb{E}_{z \sim \mu}[\ell(h, z)]$.
- The empirical risk $R_m(h) = \frac{1}{m} \sum_{i=1}^{m} \ell(h, z_i)$.

**What is PAC-Bayes learning?** PAC-Bayes learning is about learning a meaningful data-dependent posterior distribution $Q$ from a (classically data-free) prior $P$ without necessarily exploiting the Bayes formula. PAC-Bayes learning controls the *expected generalisation error*:

$$\mathbb{E}_{h \sim Q}[R(h)] := \mathbb{E}_{h \sim Q} \mathbb{E}_{z \sim \mu}[\ell(h, z)],$$

which is the averaged error that would make a predictor drawn from our posterior distribution on a new point (usually this objective holds under an iid assumption on our dataset).

**Two classical theorems.** We state below two celebrated PAC-Bayesian results: the McAllester bound enriched with Maurer's remark as stated in [Guedj, 2019, Thm.1] and Catoni's bound (Catoni, 2007, Thm 1.2.6). Those theorems both holds with the assumptions of iid data and loss bounded by 1. Note that in Thm. B.1, we also proposed another bound which is a corollary of Catoni's one.

**Theorem A.3** (McAllester' bound). *For any prior distribution $P$, we have with probability $1 - \delta$ over the $m$-sample $S$, for any posterior distribution $Q$ such that $Q \ll P$:*

$$\mathbb{E}_{h \sim Q}[R(h)] \leq \mathbb{E}_{h \sim Q}[R_m(h)] + \sqrt{\frac{KL(Q, P) + \log(2\sqrt{m}/\delta)}{2m}},$$

*where $KL$ is the Kullback-Leibler divergence.*

**Theorem A.4.** *For any prior distribution $P$, any $\lambda > 0$, we have with probability $1 - \delta$ over the $m$-sample $S$, for any posterior distribution $Q$ such that $Q \ll P$:*

$$\mathbb{E}_{h \sim Q}[R(h)] \leq \frac{1 - \exp\left\{-\frac{\lambda \mathbb{E}_{h \sim Q}[R_m(h)]}{m} - \frac{KL(Q\|P) - \log(\delta)}{m}\right\}}{1 - \exp\left(-\frac{\lambda}{m}\right)},$$

*where $KL$ is the Kullback-Leibler divergence.*

# B  Discussion about Thm. 2.3

## B.1  Comparison with classical PAC-Bayes

The goal of this section is to show how good Thm. 2.3 compared to a naive approach which consists in applying classical PAC-Bayes results sequentially. The interest of this section is twofold:

- First, presenting a classical PAC-Bayes result extracted and adapted from Alquier et al. [2016] which is formally close to what we propose.
- Second, showing that a naive (yet natural) approach to obtain online PAC-Bayes bound leads to a deteriorated bound.

We first state our PAC-Bayes bound of interest.

**Theorem B.1** (Adapted from Alquier et al. [2016], Thm 4.1). *Let $S = (z_1, ..., z_m)$ be an iid sample from the same law $\mu_0$. For any data-free prior $P$, for any loss function $\ell$ bounded by $K$, any $\lambda > 0, \delta \in ]0; 1[$, one has with probability $1 - \delta$ for any posterior $Q \in \mathcal{M}_1(\mathcal{H})$:*

$$\mathbb{E}_{h \sim Q} \mathbb{E}_{z \sim \mu_0}[\ell(h, z)] \leq \frac{1}{m} \sum_{i=1}^{m} \mathbb{E}_{h \sim Q}[\ell(h, z_i)] + \frac{KL(Q\|P) + \log(1/\delta)}{\lambda} + \frac{\lambda K^2}{2m}$$

**Remark B.2.** *Two remarks about this result:*

- *Thm. B.1 is a particular case of the original theorem from Alquier et al. [2016] as we take the case of a bounded loss which implies the subgaussianity of the random variables $\ell(., z_i)$ and then allows us to recover the factor $\frac{\lambda K^2}{m}$*

- *This theorem is derived from Catoni [2007] and constitutes a good basis to compare ourselves with as it similar formally similar.*

**Naive approach**    A naive way to obtain OPB bounds is to apply $m$ times Thm. B.1 (one per data) on batches of size 1 and then summing up the associated bounds. Thus one has the benefits of classical PAC-Bayes bound without having no more the need of data-free priors nor the iid assumption. The associated result is stated below:

**Theorem B.3.** *For any distributions $\mu_1, ..., \mu_m$ over $\mathcal{Z}$ (such that $z_i \sim \mu_i$), any $\lambda > 0$ and any online predictive sequence (used as priors) $(P_i)$, the following holds with probability $1 - \delta$ over the sample $S \sim \mu$ for any posterior sequence $(Q_i)$ :*

$$\sum_{i=1}^{m} \mathbb{E}_{h_i \sim Q_i} \left[ \mathbb{E}_{z_i \sim \mu_i}[\ell(h_i, z_i)] \right] \leq \sum_{i=1}^{m} \mathbb{E}_{h_i \sim Q_i} \left[ \ell(h_i, z_i) \right] + \frac{\mathrm{KL}(Q_i \| P_i)}{\lambda} + \frac{\lambda m K^2}{2} + \frac{m \log(m/\delta)}{\lambda}.$$

Recall that here again we assimilate the stochastic kernels $Q_i, P_i$ to the data-dependent distributions $Q_i(S, .), P_i(S, .)$

*Proof.* First of all, for any $i$, we apply Thm. B.1 $m$ to the batch $\{z_i\}$. This allows us to consider $P_i$ as a prior as it does not depend on the current data. We then have, taking $\delta' = \delta/m$, for any $i \in \{1..m\}$ with probability $1 - \delta/m$:

$$\mathbb{E}_{h_i \sim Q_i} \left[ \mathbb{E}_{z_i \sim \mu_i}[\ell(h_i, z_i)] \right] \leq \mathbb{E}_{h_i \sim Q_i} \left[ \ell(h_i, z_i) \right] + \frac{\mathrm{KL}(Q_i \| P_i)}{\lambda} + \frac{\lambda K^2}{2} + \frac{\log(m/\delta)}{\lambda}.$$

Then, taking an union bound on those $m$ events ensure us that with probability $1 - \delta$, for any $i \in \{1..m\}$:

$$\mathbb{E}_{h_i \sim Q_i} \left[ \mathbb{E}_{z_i \sim \mu_i}[\ell(h_i, z_i)] \right] \leq \mathbb{E}_{h_i \sim Q_i} \left[ \ell(h_i, z_i) \right] + \frac{\mathrm{KL}(Q_i \| P_i)}{\lambda} + \frac{\lambda K^2}{2} + \frac{\log(m/\delta)}{\lambda}.$$

Finally, summing those $m$ inequalities ensure us the final result with probability $1 - \delta$.

$\square$

**Comparison between Thm. 2.3 and Thm. B.3**    Three points are noticeable between those two theorems:

- First of all, the main issue with Thm. B.3 is that has a strongly deteriorated rate of $O\left(\frac{m \log(m/\delta)}{\lambda}\right)$ instead of the rate in $O\left(\frac{\log(1/\delta)}{\lambda}\right)$ proposed in Thm. 2.3. More precisely, the problem is that we do not have a sublinear bound: one cannot ensure any learning through time. This point justifies the need of the heavy machinery exploited in Thm. 2.3 proof as it allows a tighter convergence rate.

- The second point point lies in the controlled quantity on the left hand-side of the bound. Thm. B.3 controls $A := \sum_{i=1}^{m} \mathbb{E}_{h_i \sim Q_i} \left[ \mathbb{E}_{z_i \sim \mu_i}[\ell(h_i, z_i)] \right]$ instead of $B := \sum_{i=1}^{m} \mathbb{E}_{h_i \sim Q_i} \left[ \mathbb{E}[\ell(h_i, z_i) \mid \mathcal{F}_{i-1}] \right]$.
  $A$ is a less dynamic quantity than $B$ in the sense that it does not imply any evolution through time, it just considers global expectations. Doing so, $A$ does not take into account that at each time step we have acces to all te past data to predict the future, this may explain the deteriorated convergence rate. Thus $B$, which appears to be a suitable quantity to control to perform online PAC-Bayes (see Appendix B.2 for additional explanations)

- Finally, an interesting point is that in Thm. B.3 the bound, while looser, holds unformly for any posterior sequence contrary to Thm. 2.3 which holds only for a specific posterior sequence. This point will have a consequence for optimisation. We will come back later on this in Appendix B.3.

## B.2 A deeper analysis of Thm. 2.3

This section includes discussion about our proof technique and why all the assumptions made are necessary. We also propose a short discussion about the benefits and limitations of an online PAC-Bayesian framework as well as a deeper reflexion about the new term our bound introduce.

**Why do we need an online predictive sequence as priors?** This condition is fully exploited when dealing with the exponential moment $\xi_m$ in the proof (see Lemma D.2 proof). Indeed, the fact of having $P_i$ being $\mathcal{F}_{i-1}$-measurable is essential to apply conditional Fubini (Lemma D.3). Note that the condition $\forall i, P_{i-1} \ll P_i$ is not necessary as the weaker condition $\forall i, P_1 \ll P_i$ would suffice here. However, note that when we particularise our theorem, for instance if we choose in Corollary 3.1 $P_i = \hat{Q}_i$, one recovers the condition $\hat{Q}_i \ll \hat{Q}_{i+1}$ to have finite KL divergences. Hence the interest of taking directly an online predictive sequence.

**About the boundedness assumption** The only moment where we invoke the boundedness assumption is in Lemma D.2's proof where we apply the conditionnal Hoeffding lemma. This lemma actually translates that the sequence of r.v. $(\ell(., z_i)_{i=1..m}$ is *conditionally subgaussian* wrt the past i.e for any $i, h_i \in \mathcal{H}; \lambda \in \mathbb{R}$:

$$\mathbb{E}[\exp(\lambda \tilde{\ell}_i(h_i, z_i)) \mid \mathcal{F}_{i-1}] \leq \exp\left(\frac{\lambda^2 K^2}{2}\right)$$

where $\tilde{\ell}_i(h_i, z_i) = \mathbb{E}[\ell(h_i, z_i) \mid \mathcal{F}_{i-1}] - \ell(h_i, z_i)$.

This condition is the one truly involved in our heavy machinery. However, we chose to restrict ourselves to the stronger assumption of bounded loss function for the sake of clarity. However, an interesting open direction is to find whether there exists concrete classes of unbounded losses which may satisfy either conditional subgaussianity or others conditions (such as conditional Bernstein condition for instance).

**Reflections about the left hand side of Thm. 2.3.** We study in this paragraph the following term

$$B := \sum_{i=1}^{m} \mathbb{E}_{h_i \sim Q_i}\left[\mathbb{E}[\ell(h_i, z_i) \mid \mathcal{F}_{i-1}]\right]$$

has naturally arisen in our work as the right term to compare our empirical loss with to perform the conditional Hoeffding lemma. Taking a broader look, we now interpret this term as the right quantity to control if one wants to perform online PAC-Bayes learning. Indeed this term is a 'best of both world' quantity bridging PAC-Bayes and online learning:

- From the PAC-Bayes point of view one keeps the control on average (cf the conditional expectation in $B$) on a novel data drawn at each time step. This point is crucial in the PAC-Bayes literature as our posteriors are designed to generalise well to unseen data.

- From the Online Learning point of view, one keeps the control of a sequence of points generated from an online algorithm. Because an online learning algorithm generate a prediction for future points while having access to past data, the conditional expectation in $B$ translates this.

Finally this conditional expectation appears to be a good tradeoff between the classical expectation on data appearing in the PAC-Bayes literature (see e.g. Thm. B.1) and the local control that we have in online learning by only dealing with the performance of a sequence of points generated from a learning algorithm (see e.g. proposition A.2)

**About the interest of an Online PAC-Bayesian framework** The main shift our work does with classical online learning literature is that it does not consider the celebrated regret but instead focuses on $B$ which is a cumulative expected loss conditionned to the past. This shift does not invalidate our work but put some relief to hte guarantees Online PAC-Bayes learning can provide that Online Learning cannot and reversely.

- Online PAC-Bayes ensure a good potential for generalisation as it deals with the control of conditional expectation. This can be useful if one wants to deal with a periodic process for instance.

- Online Learning through the regret compares the studied sequence of predictors (typically generated from an online learning algorithm) and tries to compare it to the best fixed strategy (static regret) or the best dynamic one (dynamic regret). In this way, OL algorithms want to ensure that their predictions are closed from the optimal solution. This point is not guaranteed by our online PAC-Bayesian study.

- However the limitations of online learning can arise if the studied problem has a huge variance (for instance micro-transactions in finance). In this case those algorithms can follow an unpredictable optimisation route while PAC-Bayes still ensure a good performance on average (knowing the past) in this case.

- Finally, we want to emphasize that PAC-Bayesian learning circumvent a problem of *memoryless learning* which appears in classical OL algorithms. For instance, the OGD algorthm (see Appendix A.1) uses once a data and do not memorise it for further use. This problem does not happen in Online PAC-Bayes learning. Indeed, we take the example of the procedure Eq. (3) which generates Gibbs posterior which keep in mind the influence of past data.

### B.3 Thm. 2.3 and optimisation

In this section we discuss about the way Thm 2.2 can be thought in the framework of an optimisation process as we did in Secs. 3 and 4.

**A significant change compared to classical PAC-Bayes**  Thm. 2.3 holds 'for any posterior sequence $(Q_i)$ the following holds with probability $1 - \delta$ over the sample $S \sim \mu$' while most classical PAC-Bayesian results such that Thm. B.1 holds 'with probability $1 - \delta$ over the sample $S \sim \mu$ for any posterior $Q$'. This change is significant as our theorem does not control simultaneaously all possible sequences of posteriors but only holds for one. Thus, Thm. 2.3 has to be seen as a local or pointwise theorem and not as a global one. In classical PAC-Bayes, this local behavior is a brake on the optimisation process. But as we develop below, it is not the case in our online framework.

**Thm. 2.3 is compatible with online optimisation**  We first recall that classically, an online algorithm like OGD (see Appendix A.1) performs one optimisation step per arriving data. Thus, at time $m$, such algorithm will perform $m$ optimisation steps and generate $m$ predictors. Similarly the OPB algorithm of Eq. (1) generates $m$ distribution in $m$ time steps.

We insist on the fact that, Thm. 2.3 **and all its corollaries throughout our paper are valid for a sequence of $m$ posteriors and not only a single one.** A key point is that whatever the number $m$ of data, our theoretical guarantee wil still be valid for $m$ posterior distributions with the approximation term $\log(1/\delta)$ (and not $\log(m/\delta)$ as an union bound would provide for a classical PAC-Bayes theorem).

For this reason, given an online PAC-Bayes algorithm, Thm. 2.3 is suited for optimisation. Indeed, having a bound valid for a sequence of posteriors ensures guarantees for a single run of our OPB algorithm. This point is crucial to bridge a link with online learning as regret bounds (e.g. proposition A.2) also provide guarantees for a single sequence of predictors. In online learning however, those guarantees are mainly deterministic (because based on convex optimisation properties) but not totally: the recent work of Wintenberger [2021] proposed PAC regret bounds for its general Stochastic Online Convex Optimisation framework.

An interesting open challenge is to overcome the pointwise behavior of our theorem, for that, we need to rethought [Rivasplata et al., 2020, Thm 2.1] as this basis is pointwise itself. Given we consider a sequence of data-dependent priors one cannot apply the classical change of measure inequality to ensure guarantees holding uniformly on posterior sequences.

**A crucial point: having an explicit OPB/OPBD algorithm**  In our previous paragraph we said that our bound were suitable for optimisation given an OPB/OPBD algorithm. We now provide some precision about this point. All the procedures provided in the paper (i.e. Eq. (1), algorithm 1) take into account an update phase implying an argmin. Luckily for our procedures, this argmin is explicit:

- For the OPB algorithm of Eq. (1), the argmin is solved thanks to the variational formulation of the Gibbs posterior
- For OPBD algorithms, given the explicit choices of $\Psi$ given in Corollary 4.1, argmin becomes explicit when one has a derivable loss function.

In both cases, this explicit argmin ensure our procedure of interest generates explictly a single posterior per time step: we have a well-defined sequence of $m$ posteriors at time $m$. Doing so the guarantees of Thm. 2.3 holds for this sequence.

## C  A reminder on PAC-Bayesian disintegrated bounds

We present two PAC-Bayesian disintegrated bounds valid with data-dependent priors (i.e. any stochastic kernels).

- The first one is Th. 1) i) from Rivasplata et al. [2020] which provides a disintegrated version of Thm. 2.3.
- The second one is Thm 2. from Viallard et al. [2021] which involves Rényi divergence instead of the classical $KL$. Note that this bound has originally been stated for data-indepedent prior, which is why we revisit the proof to adapt it to the stochastic kernel framework.

**Proposition C.1** (Th 1) i) Rivasplata et al. [2020]). *Let* $P \in \mathcal{M}_1(\mathcal{S})$, $Q^0 \in \mathit{Stoch}(\mathcal{Z}^m, \mathcal{F})$. *Let* $f : \mathcal{S} \times \mathcal{H} \to \mathbb{R}$ *be any measurable function. Then for any* $Q \in \mathit{Stoch}(\mathcal{Z}^m, \mathcal{F})$ *and any* $\delta \in (0, 1)$, *with probability at least* $1 - \delta$ *over the random draw of* $S \sim P$ *and* $h \sim Q_S$, *we have:*

$$f(S, h) \leq \log\left(\frac{dQ_S}{dQ_S^0}(h)\right) + \log(\xi_m/\delta).$$

*where* $\xi_m := \int_\mathcal{S} \int_\mathcal{H} e^{f(s,h)} Q_s^0(dh) P(ds)$ *and* $\frac{dQ_S}{dQ_S^0}$ *is the Radon Nykodym derivative of* $Q_S$ *w.r.t.* $Q_S^0$.

**Proposition C.2** (Adapted from Th. 2 of Viallard et al. [2021]). *Let* $\mu \in \mathcal{M}_1(\mathcal{S})$, $Q^0 \in \mathit{Stoch}(\mathcal{Z}^m, \mathcal{F})$. *Let* $\alpha > 1$ *and* $f : \mathcal{S} \times \mathcal{H} \to \mathbb{R}^+$ *be any measurable function.*

*Then for any* $Q \in \mathit{Stoch}(\mathcal{Z}^m, \mathcal{F})$ *such that for any* $S \in \mathcal{Z}^m, Q_S >> Q_S^0$, $Q_S^0 >> Q_S$ *and any* $\delta \in (0, 1)$, *with probability at least* $1 - \delta$ *over the random draw of* $S \sim \mu$ *and* $h \sim Q_S$, *we have:*

$$\frac{\alpha}{\alpha - 1} \log(f(S, h)) \leq \frac{2\alpha - 1}{\alpha - 1} \log\frac{2}{\delta} + D_\alpha\left(Q_S \| Q_S^0\right) + \log\left(\underset{S' \sim \mu}{\mathbb{E}}\, \underset{h' \sim Q_{S'}^0}{\mathbb{E}}\, f\left(S', h'\right)^{\frac{\alpha}{\alpha-1}}\right)$$

*where* $D_\alpha(Q\|P) = \frac{1}{\alpha - 1} \log\left(\mathbb{E}\left[\mathbb{E}_{h \sim P}\left(\frac{dQ}{dP}(h)\right)^\alpha\right]\right)$ *is the Rényi diverence of order* $\alpha$.

Note that Viallard et al. original bound only stand for data-free priors and i.i.d data. However it appears their proof works with any stochastic kernel as prior and any distribution over the dataset. We propose below an adaptation of their proof below to fit with those more general assumptions.

### C.1  Proof of proposition C.2

*Proof.* For any sample $S$ and any stochastic kernel $Q$, note that $f(S, h)$ is a non-negative random variable. Hence, from Markov's inequality we have

$$\underset{h \sim Q_S}{\mathbb{P}}\left[f(S, h) \leq \frac{2}{\delta}\underset{h' \sim Q_S}{\mathbb{E}} f\left(S, h'\right)\right] \geq 1 - \frac{\delta}{2} \iff \underset{h \sim Q_S}{\mathbb{E}} \mathbb{1}\left[f(S, h) \leq \frac{2}{\delta}\underset{h' \sim Q_S}{\mathbb{E}} f\left(S, h'\right)\right] \geq 1 - \frac{\delta}{2}$$

Taking the expectation over $S \sim \mu$ to both sides of the inequality gives

$$\underset{S \sim \mu}{\mathbb{E}}\, \underset{h \sim Q_S}{\mathbb{E}} \mathbb{1}\left[f(S, h) \leq \frac{2}{\delta}\underset{h' \sim Q_S}{\mathbb{E}} f(S, h')\right] \geq 1 - \frac{\delta}{2}$$

$$\iff \underset{S \sim \mu, h \sim Q_S}{\mathbb{P}}\left[f(S, h) \leq \frac{2}{\delta}\underset{h' \sim Q_S}{\mathbb{E}} f(S, h')\right] \geq 1 - \frac{\delta}{2}.$$

Taking the logarithm to both sides of the equality and multiplying by $\frac{\alpha}{\alpha-1} > 0$, we obtain

$$\mathbb{P}_{S\sim\mu,h\sim Q_S}\left[\frac{\alpha}{\alpha-1}\log(f(S,h)) \leq \frac{\alpha}{\alpha-1}\log\left(\frac{2}{\delta}\mathbb{E}_{h'\sim Q_S}f(S,h')\right)\right] \geq 1 - \frac{\delta}{2}.$$

We develop the right side of the inequality in the indicator function and make the expectation of the hypothesis over $Q_S^0$ our "prior" stochadtic kernel appears. Indeed, because for any $S \in \mathcal{S}, Q_S >> Q_S^0$ and $Q_S^0 >> Q_S$ one can write properly $\frac{dQ_S}{dQ_S^0}$ and $\frac{dQ_S^0}{dQ_S} = \left(\frac{dQ_S}{dQ_S^0}\right)^{-1}$ the Radon-Nykodym derivatives. Thus we have

$$\frac{\alpha}{\alpha-1}\log\left(\frac{2}{\delta}\mathbb{E}_{h'\sim Q_S}f(S,h')\right)$$
$$= \frac{\alpha}{\alpha-1}\log\left(\frac{2}{\delta}\mathbb{E}_{h'\sim Q_S}\frac{dQ_S}{dQ_S^0}(h')\frac{dQ_S^0}{dQ_S}(h')f(S,h')\right)$$
$$= \frac{\alpha}{\alpha-1}\log\left(\frac{2}{\delta}\mathbb{E}_{h'\sim Q_S^0}\frac{dQ_S}{dQ_S^0}(h')f(S,h')\right).$$

Remark that $\frac{1}{r} + \frac{1}{s} = 1$ with $r = \alpha$ and $s = \frac{\alpha}{\alpha-1}$. Hence, we can apply Hölder's inequality:

$$\mathbb{E}_{h'\sim Q_S^0}\frac{dQ_S}{dQ_S^0}(h')f(S,h') \leq \left[\mathbb{E}_{h'\sim Q_S^0}\left(\frac{dQ_S}{dQ_S^0}(h')\right)^{\alpha}\right]^{\frac{1}{\alpha}}\left[\mathbb{E}_{h'\sim Q_S^0}f(S,h')^{\frac{\alpha}{\alpha-1}}\right]^{\frac{\alpha-1}{\alpha}}.$$

Then, by taking the logarithm; adding $\log\left(\frac{2}{\delta}\right)$ and multiplying by $\frac{\alpha}{\alpha-1} > 0$ to both sides of the inequality, we obtain

$$\frac{\alpha}{\alpha-1}\log\left(\frac{2}{\delta}\mathbb{E}_{h'\sim Q_S^0}\frac{dQ_S}{dQ_S^0}(h')f(S,h')\right)$$
$$\leq \frac{\alpha}{\alpha-1}\log\left(\frac{2}{\delta}\left[\mathbb{E}_{h'\sim Q_S^0}\left(\frac{dQ_S}{dQ_S^0}(h')\right)^{\alpha}\right]^{\frac{1}{\alpha}}\left[\mathbb{E}_{h'\sim Q_S^0}f(S,h')^{\frac{\alpha}{\alpha-1}}\right]^{\frac{\alpha-1}{\alpha}}\right)$$
$$= \frac{1}{\alpha-1}\log\left(\mathbb{E}_{h'\sim Q_S^0}\left[\frac{dQ_S}{dQ_S^0}(h')\right]^{\alpha}\right) + \frac{\alpha}{\alpha-1}\log\frac{2}{\delta} + \log\left(\mathbb{E}_{h'\sim Q_S^0}f(S,h')^{\frac{\alpha}{\alpha-1}}\right)$$
$$= D_{\alpha}\left(Q_S\|Q_S^0\right) + \frac{\alpha}{\alpha-1}\log\frac{2}{\delta} + \log\left(\mathbb{E}_{h'\sim Q_S^0}f(S,h')^{\frac{\alpha}{\alpha-1}}\right)$$

From this inequality, we can deduce that

$$\mathbb{P}_{S\sim\mu,h\sim Q_S}\left[\frac{\alpha}{\alpha-1}\log(f(S,h)) \leq D_{\alpha}\left(Q_S\|Q_S^0\right) + \frac{\alpha}{\alpha-1}\log\frac{2}{\delta} + \log\left(\mathbb{E}_{h'\sim Q_S^0}f(S,h')^{\frac{\alpha}{\alpha-1}}\right)\right]$$
$$\geq 1 - \frac{\delta}{2}. \quad (8)$$

Note that $\mathbb{E}_{h'\sim Q_S^0}f(S,h')^{\frac{\alpha}{\alpha-1}}$ is a non-negative random variable, hence, we apply Markov's inequality to have

$$\mathbb{P}_{S\sim\mu}\left[\mathbb{E}_{h'\sim Q_S^0}f(S,h')^{\frac{\alpha}{\alpha-1}} \leq \frac{2}{\delta}\mathbb{E}_{S'\sim\mu}\mathbb{E}_{h'\sim Q_S^0}f(S',h')^{\frac{\alpha}{\alpha-1}}\right] \geq 1 - \frac{\delta}{2}.$$

Since the inequality does not depend on the random variable $h \sim Q_S$, we have

$$\mathbb{P}_{S\sim\mu}\left[\mathbb{E}_{h'\sim Q_S^0}f(S,h')^{\frac{\alpha}{\alpha-1}}\leq\frac{2}{\delta}\mathbb{E}_{S'\sim\mu}\mathbb{E}_{h'\sim Q_S^0}f(S',h')^{\frac{\alpha}{\alpha-1}}\right]$$

$$=\mathbb{E}_{S\sim\mu}\mathbb{1}\left[\mathbb{E}_{h'\sim Q_S^0}f(S,h')^{\frac{\alpha}{\alpha-1}}\leq\frac{2}{\delta}\mathbb{E}_{S'\sim\mu}\mathbb{E}_{h'\sim Q_S^0}f(S',h')^{\frac{\alpha}{\alpha-1}}\right]$$

$$=\mathbb{E}_{S\sim\mu}\mathbb{E}_{h\sim Q_S}\mathbb{1}\left[\mathbb{E}_{h'\sim Q_S^0}f(S,h')^{\frac{\alpha}{\alpha-1}}\leq\frac{2}{\delta}\mathbb{E}_{S'\sim\mu}\mathbb{E}_{h'\sim Q_S^0}f(S',h')^{\frac{\alpha}{\alpha-1}}\right]$$

$$=\mathbb{P}_{S\sim\mu,h\sim Q_S}\left[\mathbb{E}_{h'\sim Q_S^0}f(S,h')^{\frac{\alpha}{\alpha-1}}\leq\frac{2}{\delta}\mathbb{E}_{S'\sim\mu}\mathbb{E}_{h'\sim Q_S^0}f(S',h')^{\frac{\alpha}{\alpha-1}}\right].$$

Taking the logarithm to both sides of the inequality and adding $\frac{\alpha}{\alpha-1}\log\frac{2}{\delta}$ give us

$$\mathbb{P}_{S\sim\mu,h\sim Q_S}\left[\mathbb{E}_{h'\sim Q_S^0}f(S,h')^{\frac{\alpha}{\alpha-1}}\leq\frac{2}{\delta}\mathbb{E}_{S'\sim\mu}\mathbb{E}_{h'\sim Q_S^0}f(S',h')^{\frac{\alpha}{\alpha-1}}\right]\geq1-\frac{\delta}{2}\quad\Longleftrightarrow$$

$$\mathbb{P}_{S\sim\mu,h\sim Q_S}\left[\frac{\alpha}{\alpha-1}\log\frac{2}{\delta}+\log\left(\mathbb{E}_{h'\sim Q_S^0}f(S,h')^{\frac{\alpha}{\alpha-1}}\right)\leq\right.$$
$$\left.\frac{2\alpha-1}{\alpha-1}\log\frac{2}{\delta}+\log\left(\mathbb{E}_{S'\sim\mu}\mathbb{E}_{h'\sim Q_S^0}f(S',h')^{\frac{\alpha}{\alpha-1}}\right)\right]\geq1-\frac{\delta}{2}.\quad(9)$$

Combining Equation Eq. (8) and Eq. (9) with a union bound gives us the desired result. □

## D   Proofs

### D.1   Proof of Thm. 2.3

**Background**   We first recall [Rivasplata et al., 2020, Thm 2].

**Theorem D.1.** *Let $\mu\in\mathcal{M}_1(\mathcal{S})$, $Q^0\in\mathtt{Stoch}(\mathcal{S},\mathcal{F})$. Let $k$ be a positive integer, any $A:\mathcal{S}\times\mathcal{H}\to\mathbb{R}^k$ a measurable function and $F:\mathbb{R}^k\to\mathbb{R}$ be a convex function . Then for any $Q\in\mathtt{Stoch}(\mathcal{S},\mathcal{F})$ and any $\delta\in(0,1)$, with probability at least $1-\delta$ over the random draw of $S\sim\mu$ we have*

$$F\left(Q_S\left[A_S\right]\right)\leq\mathrm{KL}\left(Q_S\|Q_S^0\right)+\log(\xi_m/\delta).$$

*where $\xi_m:=\int_{\mathcal{S}}\int_{\mathcal{H}}e^{f(s,h)}Q_s^0(dh)P(ds)$ and $Q_S[A_S]:=Q_S[A(S,.)]=\int_{\mathcal{H}}A(S,h)Q_S(dh)$.*

*Proof of Thm. 2.3.* To fully exploit the generality of Thm. D.1, we aim to design a $m$-tuple of probabilities. Thus, our predictor set of interest is $\mathcal{H}_m:=\mathcal{H}^{\otimes m}$ and then, our predictor $h$ is a tuple $(h_1,..,h_m)\in\mathcal{H}$. Throughout our study, our stochastic kernels $Q,Q^0$ will belong to the specific class $\mathcal{C}$ defined below:

$$\mathcal{C}:=\{Q\mid\exists(Q_i)_{i=1..m}\text{s.t. }\forall S,\ Q(S,.)=Q_1(S,.)\otimes...\otimes Q_m(S,.)\}.\quad(10)$$

Thus our kernels are such that conditionally to a given sample, our predictors $(h_1,...,h_m)$ are drawn independently.

We now apply Thm. D.1. To do so, we consider the following function $A:\mathcal{S}\times\mathcal{H}_m\to\mathbb{R}^2$ such that $\forall S=(z_i)_{i=1..m},h=(h_i)_{i=1..m}\in\mathcal{S}\times\mathcal{H}_m$:

$$A(S,h)=\left(\sum_{i=1}^m\mathbb{E}[\ell(h_i,z_i)\mid\mathcal{F}_{i-1}],\sum_{i=1}^m\ell(h_i,z_i)\right)$$

$A$ is indeed measurable in both of its variables. For a fixed $\lambda>0$, we set the function $F$ to be $F(x,y)=\lambda(x-y)$ .

The only thing left to set up is our stochastic kernels. To do so, let $P = (P_1, ... P_m)$ be an online predictive sequence, we then define $Q^0 \in \mathcal{C}$ (defined in Eq. (10)) s.t. for any sample $S$, $Q^0_S = P_1(S, .) \otimes ... \otimes P_m(S, .)$. We also fix $Q_1, ..., Q_m$ to be any (posterior) stochastic kernels and similarly we define the stochastic kernel $Q \in \mathcal{C}$ such that for any sample $S$, $Q(S, .) = Q_1(S, .) \otimes ... \otimes Q_m(S, .)$.

From now, we fix a dataset $S$ and, for the sake of clarity, we assimilate in what follows the stochastic kernels $Q_i, P_i$ to the data-dependent distributions $Q_i(S, .), P_i(S, .)$ (i.e. we drop the dependency in $S$).

Under those choices, one has:

$$Q_S[A_S] = \int_{h \in \mathcal{H}_m} A(S, h) Q_S(dh_1, ..., dh_m)$$
$$= \left( \int_{h \in \mathcal{H}_m} \sum_{i=1}^m \mathbb{E}[\ell(h_i, z_i) \mid \mathcal{F}_{i-1}] Q_S(dh_1, ..., dh_m), \int_{h \in \mathcal{H}_m} \sum_{i=1}^m \ell(h_i, z_i) Q_S(dh_1, ..., dh_m) \right).$$

Furthermore, $Q \in \mathcal{C}$, thus $Q_S(dh_1, ..., dh_m) = \Pi_{i=1}^m Q_i(dh_i)$ so:'

$$Q_S[A_S] = \left( \sum_{i=1}^m \mathbb{E}_{h_i \sim Q_i}[\mathbb{E}\left[\ell(h_i, z_i) \mid \mathcal{F}_{i-1}\right]], \sum_{i=1}^m \mathbb{E}_{h_i \sim Q_i}[\ell(h_i, z_i)] \right).$$

Finally:

$$F(Q_S[A_S]) = \lambda \left( \sum_{i=1}^m \mathbb{E}_{h_i \sim Q_i}[\mathbb{E}\left[\ell(h_i, z_i) \mid \mathcal{F}_{i-1}\right]] - \sum_{i=1}^m \mathbb{E}_{h_i \sim Q_i}[\ell(h_i, z_i)] \right).$$

Applying Thm. D.1 and re-organising the terms gives us with probability $1 - \delta$:

$$\sum_{i=1}^m \mathbb{E}_{h_i \sim Q_i} \left[\mathbb{E}[\ell(h_i, z_i) \mid \mathcal{F}_{i-1}]\right] \leq \sum_{i=1}^m \mathbb{E}_{h_i \sim Q_i}[\ell(h_i, z_i)] + \frac{KL(Q_S \| Q^0_S)}{\lambda} + \frac{\log(\xi_m/\delta)}{\lambda}.$$

Thus:

$$\sum_{i=1}^m \mathbb{E}_{h_i \sim Q_i} \left[\mathbb{E}[\ell(h_i, z_i) \mid \mathcal{F}_{i-1}]\right] \leq \sum_{i=1}^m \mathbb{E}_{h_i \sim Q_i}[\ell(h_i, z_i)] + \sum_{i=1}^m \frac{KL(Q_i \| P_i)}{\lambda} + \frac{\log(\xi_m/\delta)}{\lambda}. \quad (11)$$

The last line holding because for a fixed $S$, $Q_S = Q_1 \otimes ... \otimes Q_m$ and $Q^0_S = P_1 \otimes ... \otimes P_m$.

The last term to control is

$$\xi_m = \mathbb{E}_S \left[ \mathbb{E}_{h_1,...,h_m \sim Q^0_S} \left[ \exp\left( \lambda \sum_{i=1}^m \tilde{\ell}_i(h_i, z_i) \right) \right] \right],$$

with $\tilde{\ell}_i(h_i, z_i) = \mathbb{E}[\ell(h_i, z_i) \mid \mathcal{F}_{i-1}] - \ell(h_i, z_i)$. Hence the following lemma.

**Lemma D.2.** *One has for any $m$, $\xi_m \leq \exp\left( \frac{\lambda^2 m K^2}{2} \right)$ with $K$ bounding $\ell$.*

The proof of this lemma is deferred to Appendix D.1.1

To conclude the proof, we just bound $\xi_m$ by the result of Lemma D.2 within Eq. (11). $\qquad \square$

### D.1.1 Proof of Lemma D.2

*Proof of Lemma D.2.* We prove our result by recursion: for $m = 1$, $S = z_1$ and one knows that $P_1$ is $\mathcal{F}_0$ measurable yet it does not depend on $S$. Thus for any $h_1 \in \mathcal{H}$, $\mathbb{E}[\ell(h_1, z_1) \mid \mathcal{F}_0] = \mathbb{E}[\ell(h_1, z_1)]$.

We then has:

$$\begin{aligned}
\xi_1 &= \mathbb{E}_S \mathbb{E}_{h_1 \sim P_1}[\tilde{\ell}_1(h_1, z_1)] \\
&= \mathbb{E}_{h_1 \sim P_1} \mathbb{E}_S[\tilde{\ell}_1(h_1, z_1)] \qquad\qquad \text{by Fubini} \\
&\leq \exp \frac{\lambda^2 K^2}{2}
\end{aligned}$$

The last line holding because for any $h_1 \in \mathcal{H}$, $\tilde{\ell}_1(h_1, z_1)$ is a centered variable belonging in $[-K, K]$ a.s. and so one can apply Hoeffding's lemma to conclude.

Assume the result is true at rank $m - 1 \geq 0$. We then has to prove the result at rank $m$. Our strategy consists in conditioning by $\mathcal{F}_{m-1}$ within the expectation over $S$:

$$\xi_m = \mathbb{E}_S \left[ \mathbb{E}_{h_1, \ldots, h_m \sim Q_S^0} \left[ \exp \left( \lambda \sum_{i=1}^m \tilde{\ell}_i(h_i, z_i) \right) \right] \right].$$

First, we use that $Q^0 \in \mathcal{C}$, thus $Q_S^0 = P_1 \otimes \ldots \otimes P_m$ (i.e. our data are drawn independently for a given $S$):

$$= \mathbb{E}_S \left[ \Pi_{i=1}^m \mathbb{E}_{h_i \sim P_i} \left[ \exp \left( \lambda \tilde{\ell}_i(h_i, z_i) \right) \right] \right].$$

We now condition by $\mathcal{F}_{m-1}$ and use that $\Pi_{i=1}^{m-1} \mathbb{E}_{h_i \sim P_i} \left[ \exp \left( \lambda \tilde{\ell}_i(h_i, z_i) \right) \right]$ is a $\mathcal{F}_{m-1}$-measurable r.v.

$$\xi_m = \mathbb{E}_S \left[ \Pi_{i=1}^{m-1} \mathbb{E}_{h_i \sim P_i} \left[ \exp \left( \lambda \tilde{\ell}_i(h_i, z_i) \right) \right] \mathbb{E} \left[ \mathbb{E}_{h_m \sim P_m}[\exp(\lambda \tilde{\ell}_m(h_m, z_m))] \mid \mathcal{F}_{m-1} \right] \right].$$

Now our next step is to use a variant of Fubini valid for $\mathcal{F}_{m-1}$- measurable measures.

**Lemma D.3** (Conditional Fubini). *Let $f : \mathcal{H} \times \mathcal{Z} \to \mathbb{R}^+$. For a sigma-algebra $\mathcal{F}$ over $\mathcal{Z}$ and a measure $P$ over $\mathcal{H}$ such that*

- *$P$ is a $\mathcal{F}$-measurable r.v.*
- *There exists a constant measure (a.s.) $P_0$ such that $P >> P_0$.*

*Then one has almost surely, for any r.v. $z$ over $\mathcal{Z}$:*

$$\mathbb{E} \left[ \mathbb{E}_{h \sim P}[f(h, z)] \mid \mathcal{F} \right] = \mathbb{E}_{h \sim P} \left[ \mathbb{E}[f(h, z) \mid \mathcal{F}] \right].$$

The proof of this lemma lies at the end of this section.

We then fix $\mathcal{F} = \mathcal{F}_{m-1}$ and $f(h, z) = \exp(\lambda \tilde{\ell}_i(h, z))$. Furthermore, because we assumed the sequence $(P_i)$ to be an online predictive sequence, $P_m$ is $\mathcal{F}_{m-1}$-measurable and $P_m >> P_1$ with $P_1$ a data-free prior. One then applies Lemma D.3:

$$\mathbb{E} \left[ \mathbb{E}_{h_m \sim P_m}[\exp(\lambda \tilde{\ell}_m(h_m, z_m))] \mid \mathcal{F}_{m-1} \right] = \mathbb{E}_{h_m \sim P_m} \left[ \mathbb{E}[\exp(\lambda \tilde{\ell}_m(h_m, z_m)) \mid \mathcal{F}_{m-1}] \right].$$

Yet, injecting this result onto $\xi_m$ provides:

$$\xi_m = \mathbb{E}_S \left[ \Pi_{i=1}^{m-1} \mathbb{E}_{h_i \sim P_i} \left[ \exp \left( \lambda \tilde{\ell}_i(h_i, z_i) \right) \right] \mathbb{E}_{h_m \sim P_m} \left[ \mathbb{E}[\exp(\lambda \tilde{\ell}_i(h_m, z_m)) \mid \mathcal{F}_{m-1}] \right] \right]$$

The final remark is to notice that for any $h_m \in \mathcal{H}$, $\mathbb{E}[\tilde{\ell}_m(h_m, z_m) \mid \mathcal{F}_{m-1}] = 0$ and $\tilde{\ell}_m(h_m, z_m) \in [-K, K]$ a.s. then one can apply the conditional Hoeffding's lemma which ensure us that for any $\lambda > 0$:

$$\mathbb{E}[\exp(\lambda\tilde{\ell}_m(h_m, z_m)) \mid \mathcal{F}_{m-1}] \le \exp\left(\frac{\lambda^2 K^2}{2}\right).$$

One then has $\xi_m \le \exp\left(\frac{\lambda^2 K^2}{2}\right)\xi_{m-1}$. The recursion assumption concludes the proof.

$\square$

*Proof of Lemma D.3.* Let $A$ be a $\mathcal{F}$-measurable event. One wants to show that

$$\mathbb{E}\left[\mathbb{E}_{h\sim P}[f(h,z)]\mathbb{1}_A\right] = \mathbb{E}\left[\mathbb{E}_{h\sim P}\left[\mathbb{E}[f(h,z) \mid \mathcal{F}]\right]\mathbb{1}_A\right].$$

Where the first expectation in each term is taken over $z$. This will be enough to conclude that

$$\mathbb{E}\left[\mathbb{E}_{h\sim P}[f(h,z)] \mid \mathcal{F}\right] = \mathbb{E}_{h\sim P}\left[\mathbb{E}[f(h,z) \mid \mathcal{F}]\right]$$

thanks to the definition of conditional expectation. We first start by using the fact that $P$ is $\mathcal{F}$-measurable and that $P_0 \ll P$ with $P_0$ a constant measure. This is enough to obtain that the Radon-Nykodym derivative $\frac{dP}{dP_0}$ is a $\mathcal{F}$-measurable function, thus:

$$\mathbb{E}\left[\mathbb{E}_{h\sim P}[f(h,z)]\mathbb{1}_A\right] = \mathbb{E}\left[\mathbb{E}_{h\sim P_0}\left[f(h,z)\frac{dP}{dP_0}(h)\right]\mathbb{1}_A(z)\right],$$

$$= \mathbb{E}\left[\mathbb{E}_{h\sim P_0}\left[f(h,z)\frac{dP}{dP_0}(h)\mathbb{1}_A(z)\right]\right].$$

Because $f(h,z)\frac{dP}{dP_0}(h)\mathbb{1}_A(z)$ is a positive function, and that $P_0$ is fixed, one can apply the classical Fubini-Tonelli theorem:

$$= \mathbb{E}_{h\sim P_0}\left[\mathbb{E}\left[f(h,z)\frac{dP}{dP_0}(h)\mathbb{1}_A(z)\right]\right].$$

One now conditions by $\mathcal{F}$ and use the fact that $\frac{dP}{dP_0}, \mathbb{1}_A$ are $\mathcal{F}$-measurable:

$$= \mathbb{E}_{h\sim P_0}\left[\mathbb{E}\left[\mathbb{E}[f(h,z) \mid \mathcal{F}]\frac{dP}{dP_0}(h)\mathbb{1}_A(z)\right]\right].$$

We finally re-apply Fubini-Tonelli to re-intervert the expectations:

$$= \mathbb{E}\left[\mathbb{E}_{h\sim P_0}\left[\mathbb{E}[f(h,z) \mid \mathcal{F}]\frac{dP}{dP_0}(h)\mathbb{1}_A(z)\right]\right],$$

$$= \mathbb{E}\left[\mathbb{E}_{h\sim P}\left[\mathbb{E}[f(h,z) \mid \mathcal{F}]\mathbb{1}_A(z)\right]\right].$$

This finally proves the announced results, yet concludes the proof.

$\square$

## D.2 Proofs of Sec. 4

We prove here Corollary 4.1 and Corollary 4.2.

### D.2.1 Proof of Corollary 4.1

We fix $\hat{Q}, P$ to be online predictive sequences (with $\hat{Q}_1, P_1$ being data-free priors). Recall that we assimilated the stochastic kernels $\hat{Q}_i, P_i$ to the their associated data-dependent sitribution given a sample $S$ $\hat{Q}_i(S,.), P_i(S,.)$.

As in Thm. 2.3, our predictor set of interest is $\mathcal{H}_m := \mathcal{H}^{\otimes m}$ and then, our predictor $h$ is a tuple $(h_1, .., h_m) \in \mathcal{H}$. We consider the stochastic kernel $Q$ belonging to the class $\mathcal{C}$ defined in Eq. (10) such that for any $S \in \mathcal{S}, Q(S,.) = \hat{Q}_2 \otimes ... \otimes \hat{Q}_{m+1}$. Similarly one defines $Q^0 \in \mathcal{C}$ such that for any $S \in \mathcal{S}, Q^0(S,.) = P_1 \otimes ... \otimes P_m$

**Proof for $(\Psi_1, \Phi_1)$:** For $\lambda > 0$, we set our function $f$ to be for any dataset $S$ and predictor tuple $h = (h_1, ..., h_m)$,

$$f(S, h) = \lambda \left( \sum_{i=1}^{m} \mathbb{E}\left[\ell(h_i, z_i) \mid \mathcal{F}_{i-1}\right] - \sum_{i=1}^{m} \ell(h_i, z_i) \right).$$

We then apply proposition C.1 with the function $f, Q, Q^0$ defined above. One then has by dividing by $\lambda$ with probability $1 - \delta$ over $S \sim \mu$ and $h = (h_1, ..., h_m) \sim \hat{Q}_2 \otimes ... \otimes \hat{Q}_{m+1}$:

$$\sum_{i=1}^{m} \mathbb{E}[\ell(h_i, z_i) \mid \mathcal{F}_{i-1}] \leq \sum_{i=1}^{m} \ell(h_i, z_i) + \frac{1}{\lambda} \log \left( \frac{dQ_S}{dQ_S^0}(h_i) \right) + \frac{1}{\lambda} \log(\xi_m) + \frac{\log(1/\delta)}{\lambda}.$$

And then using the fact that $S \in \mathcal{S}, Q_S = \hat{Q}_2 \otimes ... \otimes \hat{Q}_{m+1}, Q_S^0 = P_1 \otimes ... \otimes P_m$ gives us:

$$\sum_{i=1}^{m} \mathbb{E}[\ell(h_i, z_i) \mid \mathcal{F}_{i-1}] \leq \sum_{i=1}^{m} \ell(h_i, z_i) + \frac{1}{\lambda} \sum_{i=1}^{m} \log \left( \frac{d\hat{Q}_{i+1}}{dP_i}(h_i) \right) + \frac{1}{\lambda} \log(\xi_m) + \frac{\log(1/\delta)}{\lambda},$$

with $\xi_m = \mathbb{E}_S \left[ \mathbb{E}_{h_1,...,h_m \sim Q_S} \left[ \exp \left( \lambda \sum_{i=1}^{m} \tilde{\ell}_i(h_i, z_i) \right) \right] \right]$ and for any $i$, $\tilde{\ell}_i(h_i, z_i) = \mathbb{E}\left[\ell(h_i, z_i) \mid \mathcal{F}_{i-1}\right] - \ell(h_i, z_i)$

Notice that, because $P$ is an online predictive sequence, then one can apply directly Lemma D.2 to conclude that $\xi_m \leq \exp \left( \frac{\lambda^2 K^2 m}{2} \right)$.

We also use [Viallard et al., 2021, Lemma 11] which derives the calculation of the disintegrated KL divergence between two Gaussians. One then has for any $i$, with $h_i = \hat{w}_{i+1} + \varepsilon_i$:

$$\log \left( \frac{d\hat{Q}_{i+1}}{dP_i}(h_i) \right) = \frac{||\hat{w}_{i+1} + \varepsilon_i - w_i^0||^2 - ||\varepsilon||^2}{2\sigma^2}.$$

Combining those facts altogether allows us to conclude.

**Proof for $(\Psi_2, \Phi_2)$:** For $\lambda > 0$, we set our function $f$ to be for any dataset $S$ and predictor tuple $(h = h_1, ..., h_m)$,

$$f(S, h) = \exp \left( \lambda \left( \sum_{i=1}^{m} \mathbb{E}\left[\ell(h_i, z_i) \mid \mathcal{F}_{i-1}\right] - \sum_{i=1}^{m} \ell(h_i, z_i) \right) \right).$$

We take $\alpha = 2$ and apply this time proposition C.2. One then has by dividing by $2\lambda$ with probability $1 - \delta$ over $S \sim \mu$ and $h = (h_1, ..., h_m) \sim \hat{Q}_2 \otimes ... \otimes \hat{Q}_{m+1}$:

$$\sum_{i=1}^{m} \mathbb{E}[\ell(h_i, z_i) \mid \mathcal{F}_{i-1}] \leq \sum_{i=1}^{m} \ell(h_i, z_i) + \frac{3}{2\lambda} \log \frac{2}{\delta} + \frac{D_2\left(Q_S \| Q_S^0\right)}{2\lambda} + \frac{1}{2\lambda} \log \left( \underbrace{\mathbb{E}_{S' \sim \mu} \mathbb{E}_{h' \sim Q_{S'}^0} f\left(S', h'\right)^2}_{:=\xi_m'} \right).$$

We first notice that $D_2\left(Q_S \| Q_S^0\right) = \sum_{i=1}^{m} D_2(\hat{Q}_{i+1} \| P_i)$ as our predictors are drawn independently once $S$ is given.

We also use that for any $i$, the Rényi divergence with $\alpha = 2$ between $\hat{Q}_{i+1}$ and $P_i$ (two multivariate Gaussians with same covariance matrix) is $\frac{\|\hat{w}_{i+1} - w_i^0\|^2}{\sigma^2}$ (as recalled in Gil et al. [2013]).

We then remark that:

$$\xi'_m = \mathop{\mathbb{E}}_{S'\sim\mu}\mathop{\mathbb{E}}_{h'\sim Q^0_{S'}} \exp\left(2\lambda\left(\sum_{i=1}^m \mathbb{E}\left[\ell(h'_i, z'_i) \mid \mathcal{F}_{i-1}\right] - \sum_{i=1}^m \ell(h'_i, z'_i)\right)\right).$$

Thus we recover the exponential moment $\xi_m$ from the Rivasplata's case up to a factor 2 within the exponential. We then apply Lemma D.2 with $\lambda' = 2\lambda$ to obtain that $\xi'_m \leq \exp\left(2\lambda^2 K^2 m\right)$.

Combining all those facts allows us to conclude.

### D.2.2 Proof of Corollary 4.2

We apply the exact same proof than Corollary 4.1. The only difference is the way to define our stochastic kernels. We now take, for a single online predictive sequence $\hat{Q}$ the following stochastic kernels:

We consider the stochastic kernel $Q$ belonging to the class $\mathcal{C}$ defined in Eq. (10) such that for any $S \in \mathcal{S}, Q(S, .) = \hat{Q}_1 \otimes ... \otimes \hat{Q}_m$ and we take $Q_0 = Q$.

This fact allows the divergence terms (Rényi or KL depending on which bound we consider) to vanish. The rest of the proof remains unchanged.

## E  Additional experiment

In this section we perform error bars for our OPBD methods in order to evaluate their volatility. We ran $n = 50$ times our algorithms and then show in the table below for each data set the means and the standard deviation of our averaged cumulative losses at regular time steps. We denote for $i \in \{1, 2\}$ 'OPBD $\Psi_i$' to indicate that this algorithm is our OPBD method used with thev optimisation objective $\Psi_i$.

**Analysis**  Those tables shows the robustness of our OPBD methods to their intrinsic randomness: we always have a decreasing mean through time as well as an overall variance reduction. Note that for the most complicated problem (California Housing dataset), the variance is the highest. More precisely, we notice that the standard deviation of OPBD with $\Psi_1$ is always greater than the one of OPBD with $\Psi_2$ which is not a surprise as $\Psi_1$ involves a disintegrated KL divergence while $\Psi_2$ is a proper Rényi divergence. Hence the additional volatility for OPBD with $\Psi_1$.

This fact is particulaly noticeable on the California Housing dataset where both the means and variance of OPBD with $\Psi_1$ increase drastically between t=16000 and t=20000 while the increase is more attenuated for OPBD with $\Psi_2$. This fact is also visible on Fig. 1.

|        | means OPBD $\Psi_1$ | std OPBD $\Psi_1$ | means OPBD $\Psi_2$ | std OPBD $\Psi_2$ |
|--------|---------------------|-------------------|---------------------|-------------------|
| t=200  | 0.2014              | 0.0034            | 0.1993              | 0.0007            |
| t=400  | 0.1888              | 0.0030            | 0.1861              | 0.0004            |
| t=600  | 0.1867              | 0.0023            | 0.1839              | 0.0003            |
| t=800  | 0.1714              | 0.0020            | 0.1686              | 0.0003            |
| t=1000 | 0.1760              | 0.0016            | 0.1731              | 0.0003            |

Table 1: Error bars for the Boston Housing dataset

|        | means OPBD $\Psi_1$ | std OPBD $\Psi_1$ | means OPBD $\Psi_2$ | std OPBD $\Psi_2$ |
|--------|---------------------|-------------------|---------------------|-------------------|
| t=100  | 0.1619              | 0.0063            | 0.1601              | 0.0030            |
| t=200  | 0.1350              | 0.0057            | 0.1361              | 0.0008            |
| t=300  | 0.1214              | 0.0044            | 0.1241              | 0.0009            |
| t=400  | 0.1210              | 0.0043            | 0.1238              | 0.0021            |
| t=500  | 0.1131              | 0.0037            | 0.1159              | 0.0015            |

Table 2: Error bars for the Breast Cancer dataset

|        | means OPBD $\Psi_1$ | std OPBD $\Psi_1$ | means OPBD $\Psi_2$ | std OPBD $\Psi_2$ |
|--------|---------------------|-------------------|---------------------|-------------------|
| t=150  | 0.7102              | 0.0061            | 0.7069              | 0.0007            |
| t=300  | 0.6455              | 0.0056            | 0.6422              | 0.0007            |
| t=450  | 0.6134              | 0.0042            | 0.6103              | 0.0007            |
| t=600  | 0.5860              | 0.0035            | 0.5837              | 0.0008            |
| t=750  | 0.5685              | 0.0031            | 0.5664              | 0.0008            |

Table 3: Error bars for the PIMA Indians dataset

|         | means OPBD $\Psi_1$ | std OPBD $\Psi_1$ | means OPBD $\Psi_2$ | std OPBD $\Psi_2$ |
|---------|---------------------|-------------------|---------------------|-------------------|
| t=4000  | 0.9320              | 0.0572            | 0.8905              | 0.0003            |
| t=8000  | 0.6325              | 0.0335            | 0.5947              | 0.0003            |
| t=12000 | 0.5314              | 0.0254            | 0.4954              | 0.0002            |
| t=16000 | 0.4967              | 0.0299            | 0.4477              | 0.0004            |
| t=20000 | 0.5273              | 0.1056            | 0.4355              | 0.0030            |

Table 4: Error bars for the California Housing dataset