# OpenReview forum: "Online PAC-Bayes Learning"
_NeurIPS.cc/2022/Conference — NeurIPS 2022 Accept_

### Official Review · Reviewer_UW2Y · 2022-06-21

**Rating:** 6
**Confidence:** 3
**Soundness:** 4 excellent
**Presentation:** 3 good
**Contribution:** 2 fair

**Summary:**

Traditional PAC-Bayes analysis and algorithm design is centered around a single (prior, posterior) distribution pair and a batch of typically iid data. In this work, the authors develop the machinery for analyzing learning algorithms that are characterized by a *sequence* of priors and posteriors, where data need not be iid and can be processed one by one, i.e., an "online" PAC-Bayes scenario.

The substantive content of this paper includes a general-purpose error bound that can be used in the scenario described above, and applications of this bound to derive learning algorithms. Essentially, if we have a sequence of $m$ random data points, there will be $m$ ideal objective functions of interest (i.e., one expected loss for each random data point), and the main technical result of this work bounds the sum of these "Gibbs risks" (taking expectation of each expected loss WRT the random draw from the corresponding posterior) in a typical PAC-Bayes style, such that we have a high probability bound (over the draw of the data), which holds uniformly in the choice of the posteriors. The form of the upper bound is essentially the sum of $m$ traditional PAC-Bayes bounds, except that the empirical risk term for each bound is based on a single point, and thus it is possible to derive Gibbs posteriors that are optimal in the sense of minimizing the terms in the upper bound. The authors consider several specialized settings of priors, posteriors, and distribution families, drawing links between the procedures derived from their framework and existing techniques.

**Questions:**

Why does the bounded loss assumption arise, and what consequences are there to weakening this assumption? Considering my previous comments about naively stacking up traditional PB bounds, I would like to know if there is something inherent in allowing for data-dependent priors that means we need to naively bound a sum of (squared?) losses by $mK^{2}$.

**Update:** the authors' response and revision has shed light on several key questions I had, and I have revised my score.

**Limitations:**

The authors are quite upfront about the limitations of their current approach (cumulative losses rather than regret, bounded losses, etc.) which is great, but as I mentioned earlier, there is virtually no information about why these limitations arise.

**Strengths And Weaknesses:**

Overall, the writing is quite clear, the paper is well-structured and it is easy to parse the main results of this work. The claims by the authors are sound, as they do not oversell their main result, and are quite thorough in their comparison with the existing literature.

Since the idea of an online variant of PAC-Bayes itself is not particularly novel, the novelty and value of this paper should be in the execution, i.e., the means by which the authors obtain their general purpose online PAC-Bayes bound. However, the key ideas underlying their approach are completely swept under the carpet in the main paper. All the reader has to go on is a brief comment after Thm 2.2 telling us that the authors made use of a "batch to online" technique, leveraging much of the existing PB machinery in a sequential fashion. In my opinion the nature of the online PB bound obtained, the assumptions made (e.g., bounded losses), and how these points relate to the techniques used are the most important and interesting elements of a work of this nature, yet in its current form, the paper just "gifts" the reader with a new bound and discusses the most direct consequences of this bound.

This makes it somewhat difficult for me to evaluate the novelty and significance of this work. The authors lucidly relate the consequences of their main bound to the existing literature, but the more fundamental question of how one should/could make an online PB framework and the merits/demerits of the current approach (set against other possible approaches, if any) feels like a sidenote here. Let me expand just a bit more on this point.

Assuming for now that we are satisfied with bounding the sum of expected losses, I feel like the bound in Thm 2.2 seriously lacks context. For example, if we assume that the priors are all data-independent, one could naively "stack" up $m$ traditional PB bounds ($m$ good events) applied to batches of size 1, sum these bounds, and then take a union bound. This would result in a $\log(m/\delta)$ term which is of course worse than the current $\log(1/\delta)$ term, but aside from the data-free priors, it is simple and yields a bound very similar to that of Thm 2.2, plus existing techniques can be easily used to deal with unbounded losses, etc. Of course, I know that data-dependent priors is a critical element of this work, but if this naive technique cannot be easily modified to allow for data-dependent priors, then I think that is a point the authors should communicate to the reader, and show how their approach alleviates this issue.

---

> ### Author Response · Authors · 2022-08-01
> **Response to Reviewer UW2Y**
>
> We thank you for your insightful review. First of all, we agree our main result statement lacked context. This is now fixed in a new appendix (Appendix B in the updated version of the document) dedicated to such discussion. More precisely in this appendix:
>
> - We compare our theorem to the naive approach consisting in stacking $m$ classical PB bounds in Appendix B.1. What appears there is that, simulataneaously, at each time $i$, with an union bound, one pays an approximation term of $\log(m/\delta)$. Thus summing up all the terms leads to a huge approximation term of $m\log(m/\delta)$ which is significantly worse than our result. Furthermore, the term on the LHS (left hand side) has also changed: the controlled term here is now $\sum\_{i=1}^m \mathbb{E}\_{h_i\sim Q_{i}}\left[ \mathbb{E}\_{z_i}[\ell(h_i,z_i)    \right] $ instead of $B:= \sum\_{i=1}^m \mathbb{E}\_{h_i\sim Q_{i}}\left[ \mathbb{E}[\ell(h_i,z_i) \mid \mathcal{F}\_{i-1}]    \right] $.
>
>
> -This change is further discussed in the appendix B.2, and also leads to a broader discussion about $B$. We explain why this term appears to be a best-of-both world quantity which allows us to exploit the flexibility of PAC-Bayes theory while remaining meaningful in an online framework.
>
> - We also discuss in Appendix B.2 about our proof technique, explaining why our assumptions (especially the boundedness one) are needed and why classical regret does not spontaneously appear and what it involves for the OPB (online PAC-Bayes) framework (e.g. this framework may not be the best for adversarial objectives as we do not compare ourselves to other strategies but can be adapted for instance for environment exploration objectives as we control the cumulative generalisation error at each time step ).
>
>
> Concerning your questions:
>
> - The discussion concerning the boundedness assumption is led in the paragraph 'About the boundedness assumption' l.659. We precise here that the assumption truly needed is a conditional subgaussianity (particularly implied by boundedness) and not actual boundedness. However for the sake of clarity and simplicity, we maintained the boundedness assumption. But an interesting open direction is to find whether there exists concrete classes of unbounded losses which may satisfy either conditional subgaussianity or others conditions (such as conditional Bernstein condition for instance).
>
> - For your second question, note that we actually do not naively bound a sum of squared loss, actually we do not consider a squared loss at all. What we do is exploiting the fact of having an online predictive sequence as priors to deal with the exponential moment (in Thm 2.2's proof). More precisely we use $m$ times the conditional Hoeffding lemma after the exploiting of a 'conditional Fubini' (see Lemma D.4) to obtain this factor of $\lambda m K^2 /2$. Note that a naive bound would have led to the term $mK$, which is significantly worse as this term is linear and cannnot be controlled by a right choice of $\lambda$. Cor 3.3. exploits this fact as an optimised choice of $\lambda$ allows us to recover a significant sublinear rate of $O(\sqrt{m})$.
>
> Finally we would like to challenge your comment that 'the idea of an online variant of PAC-Bayes itself is not particularly novel'.
> We partly agree as we cite in our paper previous work investigating online PAC-Bayes strategies, however we feel all these works do not fully exploit the online learning framework as they all focus on a single pair prior/posterior. To our knowledge, our work is the first to propose an online vision of PAC-Bayes with a sequence of pairs prior/posterior evolving through time.

---

> > ### Author Response · Authors · 2022-08-09
> > **Reminder**
> >
> > Dear reviewer,
> >
> > We hope that you have had chance to read and consider our response to your review, and that you would be able to share your thoughts with us.

---

> > ### Comment · Reviewer_UW2Y · 2022-08-09
> > **Re: Response to Reviewer UW2Y**
> >
> > I thank the authors for their detailed response, which along with the responses to the other reviewers has shed additional light on the various points of inquiry I had. I will raise my score to a weak accept.

---

> > > ### Author Response · Authors · 2022-08-10
> > > **Thank you for your response**
> > >
> > > We are happy to hear that the new version of the document has dissipated your concerns.
> > >
> > > We thank you for your time.

---

### Official Review · Reviewer_orn2 · 2022-07-04

**Rating:** 5
**Confidence:** 2
**Soundness:** 2 fair
**Presentation:** 2 fair
**Contribution:** 2 fair

**Summary:**

The paper employs the PAC-Bayesian proof methodology of Rivasplata et al. 2020 to provide bounds for online learning. To adapt the proof of Rivasplata et al. 2020 to the online learning setting, the loss is conditioned on the filtration of previous samples and the hypothesis space and space of posterior measures are replaced by cartesian producucts thereof. The moment generating function is bounded by recursively applying Hoeffding’s inequality for each i. In section 4, the paper discusses a disintegrated version of the bound for a hypotheses sampled from the posterior $Q$ instead of the expectation over $Q.$ Inspired by the presented PAC-Bayesian online learning bounds, the paper proposes an online learning algorithm which uses a Gibbs posterior that is iteratively updated. The learning method is evaluated on four simple binary classification and linear regression settings.


**Questions:**

* How would choose $\lambda$?
* All the related work I am familiar with that uses data dependent priors requires some additional assumption such as differential privacy. In corollary 3.3. You set P = Q such that the KL is 0. For instance, the differential privacy condition would not allow that. Can you explain why this is possible in your setting?


**Ethics Review Area:**

["I don’t know"]

**Limitations:**

The mentioned limitations and the fact that the bound constitutes a trivial upper bound is not discussed.
Ethical consideration or negative societal impact discussions are not necessary since the work is theoretical.

**Strengths And Weaknesses:**

**Strenghts:**
* Proofs seem to be correct
* Code provided
* Extensions of the PAC-Bayesian methodology to the online learning setting is a relevant open problem
**Weaknesses:**
* Questionable experiment methodology:
     * No error bars.
     * How the experiments were conducted and how the hyper-parameters are chosen is hardly described.
* Bound is extremely loose and diverges with m instead of converging to the empirical risk, neither of this is discussed in the paper.
* The clarity of the paper could be significantly improved
* A lot of the language in the paper is vague. First example in line 1 of the intro: ‘Batch learning is somewhat the dominant learning paradigm’
* As described in the summary above, the proof methodology is a straightforward extension of Rivasplata et al. 2020 to the online settings. * The paper has very few take aways that would make it worthwhile to read.

Here, I elaborate a bit more what I think the central problem with the paper is:
To my best understanding, the critical point in the paper is the trivial upper bound of the moment generating function (MGF) which results from applying Hoeffding’s inequality m times. The resulting term $\frac{\lambda m K^2}{2}$ grows linearly with $m$. The fact that $\lambda$ appears in the enumerator here but in the denominator of $\frac{\log(1/\delta)}{\lambda}$ makes it impossible to make both terms converge. Either of the two terms diverge. To my best understanding, the trivial upper bound of the MGF also leads to the fact that the authors can set $Q=P$. Typically, to obtain a meaningful bound on the MGF, we either have to assume that $P$ is independent of the samples or some weaked criterion such as differential privacy.

All together, this paper is clearly below the standard of what I would expect from a NeurIPS paper.

---

> ### Author Response · Authors · 2022-08-01
> **Response to Reviewer orn2**
>
> We warmly thank you for your review, we hope you will find our revised version easier to read.
>
> - Regarding your main concern: we are willing to defend that our bound in Thm 2.2 is not as loose as you suggest because the controlled quantity is not an expected loss on both $h$ and $z$ (e.g. $\mathbb{E}\_{h\sim Q}\mathbb{E}\_{z\sim \mu}[\ell(h,z)]$) as in the classical PAC-Bayes literature but a sum of losses (i.e $\sum\_{i=1}^m \mathbb{E}\_{h_i\sim Q_i}[\mathbb{E}[\ell(h_i,z_i) \mid \mathcal{F}\_{i-1}]] $).
> Furthermore note that in the right hand side we do not have the PAC-Bayesian empirical risk (which would be here $\frac{1}{m} \sum\_{i=1}^m \mathbb{E}\_{h_i\sim Q_i}[ \ell(h_i,z_i)]$) but the cumulative empirical error, which is the empirical risk multiplied by $m$. We refer to the paragraph "Analysis of the different terms in the bound" on page 3 line 101 and our novel Appendix B where we discuss all terms of the bound.
> Having a sublinear bound (such as the $O(\sqrt{m})$ derived from Cor 3.3) is meaningful in terms of learning. Indeed, to make our result formally closer to classical PAC-Bayes, we need to divide all the terms of Thm 2.2 by $m$. This allows us to recover, not only the classical empirical risk but also a convergence rate of $O(1/\sqrt{m})$ in Cor 3.3.
>
> We also aim to make clear that taking $P_i=\hat{Q}_i$ for all $i$ is not naive as it requires our posterior to depend only on the past, which is natural in online learning (OL). That is why we derived from Thm 2.2, Cor 3.1, which exploits the KL term and then suggest a learning algorithm and also Cor 3.3, the test bound which allows tighter convergence guarantees.
>
> Finally, we precise that considering cumulative errors instead of averaged ones strengthens the link with OL where sublinear rates are classical. For instance, we give in Appendix A.1, Thm A.2 which controls the regret (i.e. the difference between the cumulative loss wrt the best fixed strategy) with an optimal rate of $O(\sqrt{T})$. This notion of regret is fundamental in OL and further studied in (Shalev-Shwartz 2012 'Online Learning and Online Convex Optimization') or (Hazan 2016, 'Introduction to Online Convex Optimization')
>
> - Concerning our experiments, we agree error bars are needed to evaluate the variability of our experiments. That being said, we have performed error bars for our OPBD methods as they have the higher efficiency and the lower computational time. We reported our new results in Appendix E. We found that over 50 repetitions, variability is small and does not change our findings. We precised in l.340 of the main document that this appendix furnishes those error bars.
> Furthermore, we performed a GridSearch approach to find the most efficient hyperparameters. We also precise that we see our experiments as a numerical illustration of consistency of our new algorithms, we do not claim competing with state-of-the-art online algorithms but instead providing evidence that our methods efficiently converge.
>
>
> - We are happy to consider revising our writing style if you have specific examples of formulations which impede the reading.
>
> - Concerning our route of proof, we insist on the fact our results are not trivial corollaries of Rivasplata et al. 2020. Their theorem is a general basis which contains a general exponential moment. This moment is a critical quantity to control in PAC-Bayes theory. We show that under a boundedness assumption, in the OL setting, we succeed to bound this moment without further assumptions.
> To us, the control of this exponential moment in this setting is new and requires a careful study involving among others, a conditional version of Fubini (Lemma D.4).
> Furthermore the control of this exponential moment is reused in Cor 4.1 proof as the couple $(\Psi_2, \Phi_2)$ has been obtained from an adapted version of Viallard et al. 2021 (Prop C.2) involving again this exponential moment.
>
>
> **Questions**
>
> - About the choice of $\lambda$, we explained in the paragraph 'Influence of $\lambda$' l. 117 that it is seen as a trading parameter.
> In the context of Cor 3.1, we see $\lambda$ as a scale parameter for Eq. (1). We now precised it in blue l.133.
>
> In the context of Cor 3.3 we said in l.163 that $\lambda$ was seen as a tradeoff parameter between the approximation term $\log(1/\delta)$ and the ersatz $\frac{mK^2}{2}$.
>
> - In Cor 3.3, we set $\hat{Q}_i= P_i$ because of the nature of OL predictors. Indeed, the key point is that in the OL framework, we design a predictor at each time $i$ depending only on past data and possibly external informations. Concretely, this make $\hat{Q}_i$ to be $\mathcal{F}\_{i-1}$ measurable. This fact, coupled to successive absolute continuities, make $(\hat{Q}_i)_i$ an online predictive sequence and then can be used as priors.

---

> > ### Comment · Reviewer_orn2 · 2022-08-03
> > **Still strong doubts about the results in the paper**
> >
> > Thanks for your response and for extending / revising the manuscript in responds to my questions / concerns. Unfortunately, my concerns about the vacuousness and non-consistency of the bound still remain:
> >
> > Of course, I am aware that Corollary 3.1 and 3.3. constitute bounds on the cumulative loss as opposed to the average loss. However, in the same way $\frac{\lambda mK^2}{2}$ grows linearly and not sub-linearly with $m$. Thus, to my best assessment, the claim above that it would be sub-linear, i.e. $\mathcal{O}(\sqrt{m})$ is plainly wrong. When we divide both side of Corollary 3.1 or 3.3. by $m$ as suggested above by the authors, we get bounds on the average loss that are not consistent since $\frac{\lambda K^2}{2}$ stays constant with $m$. For a proper, consistent bound, we would expect that the terms vanish with $m$.
> > Let me give an example: Let's say $l$ is bounded by $K = 2$. Then we have $\frac{\lambda K^2}{2} = 2 \lambda$. Let's also assume for simplicity that the KL is 2. Then the version of Corollary 3.1. when we divide by both sides by m, bounds the generalization gap, i.e. the difference between expected average loss and empirical average loss, by a value greater than 2 which is worse than the trivial upper bound of 2 which immeditely follows from the boundedness assumption of $l \leq K = 2$.
> >
> > Moreover, what has been done in Appendix B is not convincing: Applying theorem 4.1. of Alquier et a. (2016) to each individual sample under boundedness / Hoeffding assumption and then using a union bound over the samples makes obviously little sense and yields super loose guarantees, much worse than any trivial bound that directly follows from the boundedness assumption. The strong similarity of Corollary 3.1 to this naive bound illustrates the fundamental flaws of the bound in Corollary 3, rather than supporting the validity of Corollary 3.1. I understand that you added the respective appendix section in response to one of the other reviews and am sorry for the work you put in.
> >
> > Overall, I appreciate the authors' response and efforts. However, my concerns remain and after going over the results once more I am more convinced than ever that the bounds are 1) extremely loose/vacuous 2) not consistent 3) in many cases, uniformly over all $m=1, ..., \infty$, worse than trivial bounds based on $l < K$. From my side, still a clear reject.

---

> > > ### Author Response · Authors · 2022-08-03
> > > **The answer to your main concern lies in the choice of lambda**
> > >
> > > We thank you for your response. We are willing to insist again on the convergence of our results.
> > >
> > > In the exemple you precised below (thank you for giving this specific instance), what you did not do is choosing a specific $\lambda$. We first precise the question of the choice of $\lambda$ is important in PAC-Bayes and has been treated for instance in Catoni 2007, Sec 1.2.2, Sec 1.3.1, Guedj 2019 p. 12, Audibert 2010, Sec 2.2 .
> > >
> > >  In the framework of Corollary 3.3, optimising the right hand side of the bound in $\lambda$ gives $\lambda= \sqrt{\frac{2\log(1/\delta)}{mK^2} }$. Thus we have:
> > >
> > > $ \frac{\lambda m K^2}{2} + \frac{\log(1/\delta)}{\lambda} = 2\sqrt{\frac{\log(1/\delta)mK^2}{2}}$
> > >
> > > We are allowed to take such $\lambda$ as it only depends on hyperparameters $m,\delta$. This is how we recover a sublinear rate here.
> > >
> > > We appreciate this reasoning is mainly specific to the PAC-Bayes literature and are happy to provide any additional information. We apologise about not precising it earlier and will make it more explicit on the final version.
> > >
> > > Concerning our Appendix B.1, we agree with Rev. UW2Y that putting this result is relevant because **taking $m$ batches of size $1$ in classical PAC-Bayes allows us to take history-dependent priors and non-iid data**. This point is not allowed by classical PAC-Bayes for batches of $m$ data and is exactly what we propose. Thus this approach (which deals with data sequentially), is to us the only naive one using classical PAC-Bayes theorems which allows to recover our assumption.
> > > Furthermore, the resulting bound shows that this naive approach has a clear deteriorated rate as the approximation term $\log(1/\delta)$ in Thm 2.2 has been transformed in $m\log(m/\delta)$. This bound has a linear term, even if we kill the KL terms and optimise wrt $\lambda$ (as we did above for Corollary 3.3) given the extra factor $m$. This is why we do not agree with the claim that "the strong similarity of Corollary 3.1 to this naive bound illustrates the fundamental flaws of the bound". On the contrary we argue that our bound clearly upgrades this naive bound and **allows us to obtain a sublinear convergence rate** (impossible with the naive approach). This shows it is relevant to leverage the heavy machinery used in Thm 2.2's proof.
> > >
> > > We are happy to answer any other questions you might have and we very much hope that this part of our work is now clarified. This is very helpful as we will revise the manuscript to improve the clarity of the arguments here.
> > >
> > > References:
> > >
> > > Catoni 2007, Pac-Bayesian Supervised Classification: The Thermodynamics of Statistical Learning.  (https://arxiv.org/abs/0712.0248)
> > >
> > > Guedj 2019, A primer on PAC-Bayesian Learning. (https://arxiv.org/abs/1901.05353)
> > >
> > > Audibert 2010, Agrégation PAC-Bayésienne et bandits à plusieurs bras. (https://fdocuments.net/document/agrgation-pac-baysienne-et-bandits-plusieurs-bras-pac-audibertmes-articleshdrpdf.html?page=13 )

---

> > > > ### Comment · Reviewer_orn2 · 2022-08-03
> > > > **Revised Assessment**
> > > >
> > > > I think I am wrong and you are right. Thanks for the clarifications and apologies for the confusion. I recommend that you mention that exact example of $\lambda$ in the paper as an example and state the resulting bound as a Corollary. I think it would make things much clearer + you seem to have enough space to do that.
> > > >
> > > > I am still a bit confused which is why I will set my confidence to 2 - I am clearly not on top of things here. I'll adjust my score to a 5 since the bound seems to be consistent and most likely correct. Still only a borderline accept since 1) my criticism about the experiments from above remains and 2) the clarity of the paper is still not where it should be for a NeurIPS publication.

---

> > > > > ### Author Response · Authors · 2022-08-03
> > > > > **Thanks for your revised assessment**
> > > > >
> > > > > We warmly thank you for your response and for kindly revising your assessment. We very much appreciate the opportunity for clarification and your reactivity.
> > > > >
> > > > > Regarding your experimental concerns: we feel we have answered both the points you raise in your initial review, in the rebuttal above:
> > > > >
> > > > > "No error bars": see the discussion in the first rebuttal and see Appendix E of the revised manuscript for the adding of table gathering means and variance of our cumulative loss on 50 runs of our OPBD. Furthermore all our hyperparameters have been introduced in the Parameter Settings paragraph (lines 317-327 of the revised manuscript), with brief justification when those parameters were set to specific values.
> > > > > We also precised in our rebuttal that our $\lambda$ has been chosen through a GridSearch approach. We precise this method is not novel in the PAC-Bayes literature as it is used in Theorem 3 of Mhameddi et al. 2019 which also suggest this method to chose an efficient $\lambda$.
> > > > >
> > > > > Does this answer your concerns?
> > > > > We are happy to answer any further questions you might have.
> > > > >
> > > > >
> > > > > Mhameddi et al. 2019 'PAC-Bayes Un-Expected Bernstein Inequality (https://proceedings.neurips.cc/paper/2019/file/3dea6b598a16b334a53145e78701fa87-Paper.pdf)

---

### Official Review · Reviewer_Kjzo · 2022-07-09

**Rating:** 7
**Confidence:** 3
**Soundness:** 4 excellent
**Presentation:** 3 good
**Contribution:** 3 good

**Summary:**

The paper extends the PAC-Bayes framework to an online setting, where in each time step, the algorithm is presented with one sample and outputs a posterior.
The framework is very general, in that the data distribution can be arbitrary (e.g., history-dependent), and the per-step posteriors are allowed to depend on previous posteriors and data.

 The generalization performance is measured on the new sample given the past. The main result (Thm 2.2.)  bounds this measure generalization by an “empirical error term”  and a complexity term - effectively showing a trade-off between fitting the current sample and generalizing to the next sample.

The paper then analyzes two special cases of the main theorem.
Corollary 3.1. considers the case where the per-step posterior is based on previous data (not including the current sample) and provides guarantees for the posterior applied to the last sample.  Cor. 3.3. considers the case where the posterior at each step equals the current prior and provides guarantees for the posterior applied to the current sample.
As another special case, a disintegrated PAC-Bayes bound with isotropic Gaussian measures is proved (Cor. 4.1) and is used to derive Alg. 1. Numerical experiments show preferable performance compared to OGD.


**Questions:**

1.
In Thm 2.2., I found it hard to understand the conditions on the posterior sequence and its relation to the sample set $S$.   At first reading, the statement of Thm 2.2. seems to me as a “pointwise” bound per posterior sequence, instead of a simultaneous bound over all posteriors (as in standard PAC-Bayes bounds). This formulation means that the bound does not hold for sample ($S$) -dependent posteriors (to clarify,  by “pointwise”  I mean the statement is “for any posteriors sequences w.h.p. over the sampling of $S$, bound holds”, in contrast to “simultaneous”  where the statement should be “ w.h.p. over the sampling of $S$, for any posteriors sequences, bound holds”).  If this is the case, I believe the paper should discuss this and explain the challenges in obtaining a “simultaneous” bound in this setting.

2.
Reading the proof of the theorem, it seems that $Q_i$ is defined as stochastic kernels (Def. C.1.)
as in Rivasplata et al., 2020. This is not mentioned at all in the main body of the paper. To my understanding, the formulation in the main body differs from the one used in the proof. Shouldn't the main body of the paper use this definition?

3.
Does the algorithm derived in the paper have ensured generalization guarantees?
I.e., is the stochastic kernel that minimizes the RHS of the bound of Thm 2.2. given a dataset, has the generalization guarantees predicted by Thm 2.2.?
 In usual PAC-Bayes bounds, this is true, since the bound simultaneously holds for all posterior distributions. I wonder if the same conclusion can be drawn here.
As stated in “Rivasplata, Omar. PAC-Bayesian Computation. Diss. UCL (University College London), 2022”, Sect 3.2, the stochastic-kernel-based bound is not suitable for optimization. I have not found a discussion of this in the paper in the algorithm and experiments sections.

3.
In line 143, and in Remark 3.2. the prior is said to be $\hat{Q}_i,$ while in Eq. 1, the prior is $P_i$.  What is the reason for the discrepancy?

4. Can the authors expand on the significance of Cor. 3.3.? How is it different from applying Chernoff’s bound per step?

5.
Lines 181-191 (Analogy with Online Gradient Descent) - the claims here can be written more clearly and explicitly (e.g., can you show explicitly how the analogy holds? what is the relation between lambda and mu?)



**Limitations:**

The limitations have been discusse above.

**Strengths And Weaknesses:**

### Strengths
I believe the work is original and well-motivated.
The extension of the PAC-Bayes framework to the online setting (and the analyzed performance measure) is logical and elegant. The framework requires very limited assumptions, allowing general data distributions and history-dependent posteriors.

The paper is well-written and clear.
The authors provide an adequate survey of related work.

All in all, I think the paper has the potential to make important contributions to the  PAC-Bayes literature.

### Weaknesses
1. Regarding the experimental results, I think it would be interesting to add a comparison with the prediction of the bounds.
2.  I have a few issues (see questions below) that I would like the authors to clarify in their rebuttal.

---

> ### Author Response · Authors · 2022-08-01
> **Response to Reviewer Kjzo**
>
> We warmly thank you for your enthusiasm about our work and your insightful review. Regarding to your concerns about the links between our main theorem and optimisation, we created a new appendix (namely Appendix B.3) which addresses this topic. We also provide below a summary of this discussion.
>
> Weaknesses:
> 1. Concerning your experimental concern, the reason for which we only plotted the averaged empirical loss is that the bound of Cor 3.3 (after dividing by $m$ for averaging) consists in this averaged empirical loss plus an approximation term in $O(1/\sqrt{m})$ It appears that given our dataset sizes, the magnitude of the approximation term is at least $1/\sqrt{500}\approx 10^{-2}$ while our averaged cumulative loss is always of magnitude between 1 and $10^{-1}$. Thus, plotting those additional terms would have led to similar curves and less clear graphs.
>
> Questions:
> 1. and 3. First of all, we acknowledge we did not precise the fact that the posterior sequence can be data-dependent, we precised it in blue l.90 in the revised version.
>
> Second, it is true that Thm 2.2 is 'pointwise' in the sense you propose. We discuss this in Appendix B.3. A sum up of this discussion would be that the bound is suited for optimisation for two reasons:
> - contrary to classical PAC-Bayes, our bound holds for a sequence of posteriors with high probability.
> - the argmin is explicit (Gibbs posterior)
>
> The second point affirms that the learning algorithm derived in Eq.2 (line 137) generates explictly a single posterior per time step: we have a well-defined sequence of $m$ posteriors at time $m$. Doing so the guarantees of Thm 2.2 holds for this sequence. The main difference between our result and the original one from Rivasplata et al. is that the original stochastic has (probably) been thought for one data-dependent distribution while our results invoke a stochastic kernel generating $m$ data-dependent distribution. Doing so, our theorem, while pointwise, is pointwise for a sequence of posteriors and so still ensure guarantees for a single run of an online PAC-Bayes algorithm. Indeed, if the argmin is explicit as for the one of Eq. (1) l. 137 then our learning algorithm derives a sequence of $m$ posteriors in $m$ time steps.  To us, this point is crucial to bridge a link with online learning as regret bounds (e.g. Prop A.2) also provide guarantees for a single sequence of predictors (in prop A.2 it is the one generated by OGD).
> However to overcome the pointwise behavior of our theorem, we need to adapt Rivasplata et al. 2020 (Thm 2.1) as this basis is pointwise itself. Given we consider a sequence of data-dependent priors one cannot apply the change of measure inequality to ensure guarantees holding uniformly on posterior sequences. More discussion can be found in Appendix B.3
>
>
> 2. We technically can define any data-dependent posterior $Q_i$ (and even any prior $P_i$) as a stochastic kernel. We did not do this choice for two reasons:
> - The first one is clarity. We believe that adding the definition of stochastic kernel to refer to data-dependent measure would have add confusion to the reader non-familiar with the work of Rivasplata et al. 2020. It is why we only evoke data-dependent measures in the main document as we believe it is more easier to understand for non PAC-Bayesian specialists (we also want to reach the online learning community with this work).
> - The second one is about clarity. In Thm 2.2's proof, we generate a stochastic kernel $Q(.,S)$ taking into account the $m$ data-dependent posteriors $Q_i(S)$. We think it would be easier to understand proof if were only using two stochastic kernels $P,Q$ as in the original theorem of Rivasplata et al. 2020.
>
> 4. The reason of this discrepancy is precised in the paragraph 'Strength of our results' l. 205. We say that taking $P_i= \hat{Q}_i$ is only a particular of what Cor. 3.1 can provide. We illustrate this point with the following example: 'if our online predicitve sequence $(\hat{Q}_i)$ can be defined through a sequence of parameter vectors $\hat{\mu}$, then we can define $P_i$ by adding a small noise on $\hat{\mu}_i$ and thus giving more freedom through stochasticity.'
>
> 5. We presented Cor 3.3 as a OPB test bound. We acknowledge that, because we killed the KL divergence term, this result is not specifically PAC-Bayesian in itself and is potentially reachable with other routes of proofs as the one you propose. However we stress that Cor 3.3 is only one side of Thm 2.2 as Cor 3.1. So, our goal through those corollaries is to illustrate the richness of Thm 2.2 by instanciating two meaningful corollaries (training and test bounds).
>
> 6. We explicited our analogy in the case where $\hat{Q}_i=\mathcal{N}(\hat{m}_i,I_d)$ modifications are written in blue on l.183. It allows us to recover explicitly the OGD algorithm for the averaged loss functions if we minimise a Taylor expansion of those losses at each time step. We then precisely show the smilar role of $\lambda$ and $\eta$.

---

> > ### Comment · Reviewer_Kjzo · 2022-08-06
> > **Regarding item 2**
> >
> > I thank the authors for their clarification and revisions.
> >
> > Regarding item 2 in the comment above: to me, it seems that formulating the main theorem (2.2) with stochastic kernels is crucial for justifying the algorithm derivation.
> > The formulation of Thm 2.2 (for any Q_i, w.h.p. over S) means that the Q_i needs to b fixed (non-data-dependent)for the guarantee to hold (correct me if I am wrong)
> > But, of course,  the algorithm presented in Eq (2) outputs a data-dependent posterior.
> > This discrepancy may be accounted for since Thm 2.2 should be formulated for stochastic kernels instead of posterior distributions.

---

> > > ### Author Response · Authors · 2022-08-08
> > > **About item 2**
> > >
> > > We thank your for this discussion.
> > >
> > > Concerning your last concern, we now agree that it is essential to introduce properly stochastic kernels in the main document to dissipate the final confusion around item 2.  Indeed, you seem to suggest is that, because the stochastic kernels $Q_i(.,.)$ (taking as first argument a sample and an event as second as in Def D.1)  are fixed before the statement of the highly probable event, then our result holds for data-free measures.
> > > This is not true for the following reason: *even if the stochastic kernel $Q_i(.,.)$ is fixed, this is not the case for the data-dependent measure $Q_{i}(S,.)$*.
> > >
> > > To emphasize this point note that fixing the stochastic kernels in our theorems is like fixing a learning algorithm $A$ which takes as input a sample $S$ and outputs a sequence of $m$ distributions $Q_1(S,.),...,Q_m(S,.)$ : the outputs of $A$ are data-dependent while $A$ in itself is not.
> > >
> > > Thus, a more rigorous statement of Theorem 2.2 is:
> > >
> > > 'For any distribution $\mu$ over $\mathcal{Z}^m$, any $\lambda>0$ and any online predictive sequence (used as priors) $(P_i)$, for any sequence of stochastic kernels $(Q_i)$ we have with probability $1-\delta$ over the sample $S\sim\mu$ the following, which holds for the data-dependent measures $Q_1(S,.),...,Q_m(S,.)$:
> > > ...
> > > '
> > >
> > > the bound remains unchanged with the shift of notations $Q_i\rightarrow Q_i(S,.)$.
> > >
> > >
> > > What we propose to do in the next version is:
> > >
> > > - update the definition of online predictive sequence by also incorporating the notion of stochastic kernel
> > > - introducing properly stochastic kernels in the main document
> > > - modifying the statement of our theorems as precised below
> > > - adding a short paragraph which explain stochastic kernels to non-specialists.
> > >
> > > Does this address your final concern?

---

> > > > ### Comment · Reviewer_Kjzo · 2022-08-09
> > > > **respnse to authors**
> > > >
> > > > I thank the authors for the detailed response, it completely addresses my concern.

---

### Official Review · Reviewer_zWwp · 2022-07-11

**Rating:** 6
**Confidence:** 2
**Soundness:** 3 good
**Presentation:** 1 poor
**Contribution:** 3 good

**Summary:**

This paper extends the classical PAC-bayes theory to the online learning setting, where the sequence of priors and posteriors can dynamically involve. The authors provide the first online version of training and testing bound, and drive OGD-like algorithm for this problem.

**Questions:**

The questions are listed above.

**Ethics Review Area:**

["I don’t know"]

**Limitations:**

Yes.

**Strengths And Weaknesses:**

Strengths:

1.	This paper considers PAC-Baysian learning in an online fashion, which is a natural extension of the classical PAC-Bayes results and to my knowledge is novel.
2.	The main contribution in Theorem 2.2 seems to be very general, and even holds for non-convex bounded functions.
3.	Experimental results demonstrate the effectiveness of the proposed methods.

Weaknesses:

1. I find the paper is difficult to read especially for readers who are not very familiar with PAC-Bayes theory. I think a preliminary on the classical PAC-Bayes before introducing the online version in Section 2 would be super helpful, and a comparison between Theorem 2.2 and the classical PAC-Bayes theory is also needed. Moreover, could the authors provide the definition of absolute continuity (line 88)?


2. I have the following questions:

Can the authors provide more details about the LHS of the inequality in Theorem 2.2? More specifically, about the expectation inside (the expectation is taken wrt which random variables)?

If the data is iid, does Theorem 2.2 reduces to the classical conclusions?

Why corollary 3.1 suggests the algorithm in (1), if we perform the algorithm in (1), are their any follow up results (like regret)? And how do one compute the expectation in (1) in practice? (Line 181) Compared to OGD, is the algorithm more related to OMD or exponential gradient?

It seems to me that Corollary 3.1 and 3.2 are just Theorem 2.2 with different notations, and I’m not sure if it is necessary to repeat the same conclusion for several times.

Could the authors be more clear in the introduction about the contributions of Section 4?

---

> ### Author Response · Authors · 2022-08-01
> **Response to Reviewer zWwp**
>
> We warmly thank you for your positive review.
>
> First of all we hope you will find our revised version clearer and easier to read. We added in appendix A a reminder about PAC-Bayes learning with some classical theorems. Note that, in order to provide a broader vision of our work, according to other reviewer's remarks, we also created Appendix B which provides many discussions about Thm 2.2 and we added the definition of absolute continuity l.77. We hope those two appendices will provide you a broader vision of our work and what PAC-Bayes does.
>
> Concerning your questions:
>
> - We provided a broader discussion of the term in the LHS in the paragraph 'Reflections about the left hand side of Thm 2.2' (App B.2, l. 668). Furthermore, we precised l.78-79 that the filtration was adapted to the sample $S$ which implies the conditional expectation $\mathbb{E}[\ell(h_i,z_i) \mid \mathcal{F}_{i-1}]$ holds wrt $z_i$ for any $i$.
>
> - If the data is iid, Thm 2.2 does not recover classical guarantees of PAC-Bayes learning. The reason is that we dealt with data sequentially, which implies some additional factor compared to the batch approach.
> However, there exists a more meaningful way to compare ourselves to classical PAC-Bayes (as suggested by another reviewer): what if we treat sequentially our data in classical PAC-Bayes? How tight this bound would be? It appears this approach leads to a much looser bound than Thm 2.2, this highlights the strength of our results. This approach is presented in Appendix B.1 l.598 along with additional discussions.
>
> - Cor 3.1 suggests the algorithm presented in Eq. (1) l.137 because its RHS provides an empirical surrogate of the LHS of Thm 2.2. Thus, minimising this upper bound allow us to obtain an numerical upper bound on the LHS. Doing so, our learning objectives are derived to ensure measurable upper bounds on the CGE. This approach is the basic way to derive PAC-Bayes algorithms.
> Furthermore, our approach does not provide regret bounds but once we ran a single run of a online PAC-Bayes algorithms (or its disintegrated counterpart in Sec.4), we still have access to the guarantees provided by Cor 3.3 and Cor 4.2 for our posterior sequence.
>
> - About the computation of the expectation in Eq. 1, we ran an MCMC approach to estimate this moment. We acknowledge this may have a huge time complexity. This is why we proposed Sec. 4 which provides disintegrated PAC-Bayesian algorithms which do not need to estimate an expectation.
>
> - To us, our algorithms are more related to OMD as we see our KL term as the analoguous of the regularisation function appearing in OMD.
>
> - Technically speaking both Cor 3.1 and 3.3 are immediate consequences of Thm 2.2. However, we emphasise those bounds have two completely different goals. Cor 3.1 is necessary to derive online PAC-Bayes algorithms while Cor 3.3 allows us to obtain tight convergence guarantees. We insist especially on the significant difference in the LHS of both corollaries. We analysed those terms respectively in Remark 3.2 l.146 and l. 161 (added in blue).
>
> - We added in the introduction l. 53 that Sec. 4 circumvents the problem of expectation estimation that naturally appears with Gibbs posteriors in Sec.3. Disintegrated PAC-Bayes bounds allow us to obtain time efficient online algorithms.

---

### Author Response · Authors · 2022-08-01
**General response**

We warmly thank all four reviewers for their careful evaluation of our work, and we are especially encouraged by the positive elements in the reviews (in particular its potential impact on the PAC-Bayes literature, its clarity and presentation, and main results). We provide a point-by-point response to each review below and we hope reviewers with less positive evaluations will be convinced of the merits of our work.

---

### Comment · Area_Chair_4out · 2022-08-03
**Additional question**

Dear authors,

An additional question was raised during the discussion with the reviewers, so I think it is fair to give you the opportunity to reply:

Theorem 3 of http://proceedings.mlr.press/v75/hoeven18a/hoeven18a.pdf proposes a Gaussian approximation of the exponential weights with fixed variance, as in your Section 4, and a linearisation leads to the online gradient algorithm. Could you compare your regret bound to theirs? [by the way, I think it would be fair to cite their paper] Also, what would be your interpretation on the empirical performance of your method versus OGD? According to Hoeven et al, we would expect them to be quite similar...

---

### Meta-Review · Area_Chair_4out · 2022-08-20

**Recommendation:** Accept
**Confidence:** Less certain

**Metareview:**

PAC-Bayes theory provides upper-bounds on the risk of aggregation of predictors in the batch setting. Many PAC-Bayes bounds are actually minimized by EWA (Exponentially Weighted Aggregation), but these bounds can also be applied on (slightly) sub-optimal aggregation procedure, and allow to control their level of sub-optimality: classical examples include Gaussian aggregation / variational Bayes.

There are also bounds on the regret of EWA in the online setting. However, while these bounds look quite similar to PAC-Bayes bound, they usually do not allow to work with alternate aggregation procedure such as Gaussian aggregation. Recently, some results allowed to study other aggregation rules, as in van der Hoeven et al. [2018] and Chérief-Abdellatif et al. [2019], but these results still impose strong constraint and cannot be applied to arbitrary aggregation strategies.

Here, the authors manage to extend totally PAC-Bayes bounds to the online setting. In other words, their Theorem 2.2 can be used to upper bound the regret of very general aggregation strategies, including of course EWA and Gaussian aggregation.

There was initially a disagreement between reviewers, based on the following:
1) on the one hand, the reviewers agree that Theorem 2.2 is a nice extension of PAC-Bayes bounds, and provides a generalization of existing results on EWA and Gaussian aggregation [see Reviewers Kjzo, orn2 and also UW2Y].
2) on the other hand, it is not clear whether there is a useful application of Theorem 2.2 can lead to new results beyond the "usual cases" EWA / Gaussian. Indeed, these are the two examples discussed by the authors [UW2Y].
After discussion, there was ultimately an agreement that even though some of the reviewers and myseld are still not totally convinced about 2), the nice construction of 1) justifies publication of the paper. I will therefore recommend to accept it.

Each of the reviewers raised many minor issues that the authors should take into account in the camera ready version (writing [zWwp, UW2Y] / experiments [Kjzo, orn2] / ...). I will add the following points:
- van der Hoeven et al. [2018] already contains a nice discussion on the extension of regret bounds beyond EWA, and share many similarities with this work (even though it does NOT contain a result such as Theorem 2.2). This paper is currently not cited by the authors. The authors should cite it, and discuss it.
- "The guarantees Chérief-Abdellatif et al. [2019] provided for SVB hold for Gaussian priors and posteriors and are valid for iid data, which is a particular case of our work." This is an incorrect and misleading statement: this paper is written in the same setting than the classical bounds on EWA. There is no stochastic assumption on the data in this paper (nor in van der Hoeven et al. [2018]).

**Award:**

No

---

### Decision · Program_Chairs · 2022-09-14

Accept